# Risk factors for prostate cancer: An umbrella review of prospective observational studies and mendelian randomization analyses

Huijie Cui[1©], Wenqiang Zhang[1©], Li Zhang[1©], Yang Qu[1], Zhengxing Xu[1], Zhixin Tan[1], Peijing Yan[1], Mingshuang Tang[1], Chao Yang[1], Yutong Wang[1], Lin Chen[1], Chenghan Xiao[2], Yanqiu Zou[1], Yunjie Liu[1], Ling Zhang[3], Yanfang Yang[1], Yuqin Yao[4], Jiayuan Li[1], Zhenmi Liu[2], Chunxia Yang[1], Xia Jiang[1,5,6]*, Ben Zhang[7]*

1 Department of Epidemiology and Biostatistics, Institute of Systems Epidemiology, and West China-PUMC C. C. Chen Institute of Health, West China School of Public Health and West China Fourth Hospital, Sichuan University, Chengdu, Sichuan, China, 2 Department of Maternal, Child and Adolescent Health, West China School of Public Health and West China Fourth Hospital, Sichuan University, Chengdu, China, 3 Department of Iatrical Polymer Material and Artificial Apparatus, School of Polymer Science and Engineering, Sichuan University, Chengdu, China, 4 Department of Occupational and Environmental Health, West China School of Public Health and West China Fourth Hospital, Sichuan University, Chengdu, China, 5 Department of Nutrition and Food Hygiene, West China School of Public Health and West China Fourth Hospital, Sichuan University, Chengdu, China, 6 Department of Clinical Neuroscience, Karolinska Institute, Stockholm, Sweden, 7 Hainan General Hospital and Hainan Affiliated Hospital, Hainan Medical University, Haikou, China; West China School of Public Health and West China Fourth Hospital, Sichuan University, Chengdu, China

© These authors contributed equally to this work.
* xiajiang@scu.edu.cn (XJ); benzhang@vip.163.com (BZ)

**Data Availability Statement:** All relevant data are within the manuscript and its Supporting information files.

## Abstract

### Background

The incidence of prostate cancer is increasing in older males globally. Age, ethnicity, and family history are identified as the well-known risk factors for prostate cancer, but few modifiable factors have been firmly established. The objective of this study was to identify and evaluate various factors modifying the risk of prostate cancer reported in meta-analyses of prospective observational studies and mendelian randomization (MR) analyses.

### Methods and findings

We searched PubMed, Embase, and Web of Science from the inception to January 10, 2022, updated on September 9, 2023, to identify meta-analyses and MR studies on prostate cancer. Eligibility criteria for meta-analyses were (1) meta-analyses including prospective observational studies or studies that declared outcome-free at baseline; (2) evaluating the factors of any category associated with prostate cancer incidence; and (3) providing effect estimates for further data synthesis. Similar criteria were applied to MR studies. Meta-analysis was repeated using the random-effects inverse-variance model with DerSimonian—Laird method. Quality assessment was then conducted for included meta-analyses using AMSTAR-2 tool and for MR studies using STROBE-MR and assumption evaluation. Subsequent evidence grading criteria for significant associations in meta-analyses contained

**Funding:** The National Natural Science Foundation of China: U22A20359, 81874283, and 81673255, granted to BZ; the National Key R&D Program of China: 2022YFC3600604, granted to BZ; the Recruitment Program for Young Professionals of China, the Promotion Plan for Basic Medical Sciences and the Development Plan for Cutting-Edge Disciplines, Sichuan University, and other Projects from West China School of Public Health and West China Fourth Hospital, Sichuan University, granted to BZ. The National Natural Science Foundation of China for young scholars: 82204170, granted to XJ; the National Natural Science Foundation of China for young outstanding scholars (overseas), granted to XJ. The sponsors or funders had no role in the study design, data collection and analysis, decision to publish, or preparation of the manuscript.

**Competing interests:** The authors have declared that no competing interests exist.

**Abbreviations:** AASV, anti-neutrophil cytoplasm antibody associated vasculitide; ACEI, angiotensin converting enzyme inhibitor; BMI, body mass index; BPH, benign prostatic hyperplasia; CCB, calcium-channel blocker; CRP, C-reactive protein; DHA, docosahexaenoic acid; FA, favorable adiposity; FGF, fibroblast growth factor; HDL, high-density lipoprotein; HR, hazard ratio; IQR, interquartile range; IRR, incidence rate ratio; IV, instrumental variable; LCI, lower confidence interval; LDL, low-density lipoprotein; LTL, leukocyte telomere length; MR, mendelian randomization; mTOR, mammalian target of rapamycin; NSAID, nonsteroidal anti inflammatory drug; OR, odds ratio; PA, physical activity; PI, prediction interval; PSA, prostate-specific antigen; RR, risk ratio; SD, standard deviation; SLE, systemic lupus erythematosus; T2D, type 2 diabetes; UC, ulcerative colitis; UCI, upper confidence interval; UFA, unfavorable adiposity.

sample size, $P$ values and 95% confidence intervals, 95% prediction intervals, heterogeneity, and publication bias, assigning 4 evidence grades (convincing, highly suggestive, suggestive, or weak). Significant associations in MR studies were graded as robust, probable, suggestive, or insufficient considering $P$ values and concordance of effect directions.

Finally, 92 selected from 411 meta-analyses and 64 selected from 118 MR studies were included after excluding the overlapping and outdated studies which were published earlier and contained fewer participants or fewer instrument variables for the same exposure. In total, 123 observational associations (45 significant and 78 null) and 145 causal associations (55 significant and 90 null) were categorized into lifestyle; diet and nutrition; anthropometric indices; biomarkers; clinical variables, diseases, and treatments; and environmental factors. Concerning evidence grading on significant associations, there were 5 highly suggestive, 36 suggestive, and 4 weak associations in meta-analyses, and 10 robust, 24 probable, 4 suggestive, and 17 insufficient causal associations in MR studies. Twenty-six overlapping factors between meta-analyses and MR studies were identified, with consistent significant effects found for physical activity (PA) (occupational PA in meta: OR = 0.87, 95% CI: 0.80, 0.94; accelerator-measured PA in MR: OR = 0.49, 95% CI: 0.33, 0.72), height (meta: OR = 1.09, 95% CI: 1.06, 1.12; MR: OR = 1.07, 95% CI: 1.01, 1.15, for aggressive prostate cancer), and smoking (current smoking in meta: OR = 0.74, 95% CI: 0.68, 0.80; smoking initiation in MR: OR = 0.91, 95% CI: 0.86, 0.97). Methodological limitation is that the evidence grading criteria could be expanded by considering more indices.

## Conclusions

In this large-scale study, we summarized the associations of various factors with prostate cancer risk and provided comparisons between observational associations by meta-analysis and genetically estimated causality by MR analyses. In the absence of convincing overlapping evidence based on the existing literature, no robust associations were identified, but some effects were observed for height, physical activity, and smoking.

### Author summary

#### Why was this study done?

- The incidence of prostate cancer is increasing with the growing trend of aging globally.

- Effective preventions and interventions for prostate cancer require better understandings of its etiology.

- The well-known risk factors for prostate cancer are age, ethnicity, and family history, but few modifiable factors have been firmly established.

**What did the researchers do and find?**

- Our study extensively collected, evaluated, and compared the current observational and genetic evidence for various factors modifying the risk of prostate cancer based on meta-analyses and mendelian randomization (MR) studies.

- Totally 123 observational associations (45 significant and 78 null) from 92 meta-analyses and 145 causal associations (55 significant and 90 null) from 64 MR studies were identified and categorized into lifestyle; diet and nutrition; anthropometric indices; biomarkers; clinical variables, diseases, and treatments; and environmental factors.

- Concerning evidence grading on significant associations, there were 5 highly suggestive, 36 suggestive, and 4 weak associations in meta-analyses, and 10 robust, 24 probable, 4 suggestive, and 17 insufficient causal associations in MR studies.

- Consistent significant associations between meta-analysis and MR studies were found for physical activity, height, and smoking, which however were not robust.

**What do these findings mean?**

- Most included cohort studies were conducted in developed western countries, and hence the findings in this study are limited for mainly European descendants.

- The comparison between observational associations by meta-analysis and genetically estimated causality by MR analyses does not provide robust evidence due to the lack of overlapping associations and high-quality evidence, especially in MR studies.

- Evidence grading criteria for meta-analyses could be further improved by adding more indices such as magnitude of effect size and different levels of sample size.

## Introduction

Prostate cancer is the second most frequent cancer and the fifth leading cause of cancer-related death among men, and its incidence is increasing in older males with the growing trend of aging globally [1]. Effective early preventions and interventions for prostate cancer require better understandings to its etiology which represents a complex interplay between genetic susceptibility and micro- and macro-environmental factors [2]. Observational studies have investigated and identified a plethora of factors associated with the risk of prostate cancer [3–5]. The well-known risk factors for prostate cancer are age, ethnicity, and family history, but few modifiable factors have been firmly established.

Umbrella review aggregates evidence from published meta-analysis and structurally summarizes evidence strength to provide an inclusive overview on a given topic via a comprehensive assessment of sample size, strength and precision of the association, heterogeneity, and biases [6–8]. The earliest umbrella review on prostate cancer, to our knowledge, was published in 2016, focusing on diet, body size, and physical activity [9]. Other existing umbrella reviews, involving prostate cancer as one of the many health outcomes, were specifically limited to dietary factors including folate [10], fish and ω-3 fatty acids [11], tomato and lycopene [12], and

whole grain consumption [13]. Several important factors including lifestyle; environmental exposures; and preexisting clinical variables, diseases, and treatments, are often overlooked by existing umbrella reviews.

In addition to observational studies, mendelian randomization (MR) studies leverages genetic variations as proxies for exposures to obtain unbiased effect estimates, minimizing the influence of reverse causation or confounding which is often found in epidemiological settings [14]. MR studies have been extensively conducted to explore potential causal risk factors for prostate cancer [15–18], part of which have been summarized and assessed in the systematic review of MR studies by Markozannes and colleagues [19], yet needing update by including newly published MR studies.

Therefore, an updated comprehensive umbrella review on prostate cancer is needed. To ensure the evidence quality from observational studies, meta-analyses of prospective observational studies are preferred as they clearly indicate temporal relationship between exposure and outcome and are thus less biased than retrospective studies [20]. Similarly, MR studies provide unbiased evidence because the genotypes are defined at conception bases on the random assortment of genes and thus not influenced by conventional confounders [21]. It could be beneficial to compare epidemiological studies informing association and MR studies suggesting causality and investigate their mutual corroboration or discrepancy, to gain mutually complementary insights on understanding the risk of prostate cancer. Therefore, the objective of this umbrella review is to identify and evaluate various factors modifying the risk of prostate cancer reported in meta-analyses of prospective observational studies and MR studies, to better understand the etiology of prostate cancer.

## Methods

### Literature search and eligibility criteria

This study is reported as per the Preferred Reporting Items for Systematic Reviews and Meta-Analyses (PRISMA) guideline (S1 PRISMA Checklist). No preregistered study protocol is available. This umbrella review was initially planned to focus on evidence from observational studies, so the initial search was conducted on January 10, 2022 only for meta-analyses. An additional search for MR studies was later conducted on July 6, 2022, to include the important genetic evidence from MR studies. Upon request, the literature search for meta-analyses and MR studies was updated on September 9, 2023.

Systematic literature search was conducted in PubMed, Embase, and Web of Science. A predefined comprehensive search strategy (S1 Text) was used to search all meta-analyses and MR studies evaluating various factors associated with prostate cancer risk from the inception of database to September 9, 2023. We also searched Cochrane Database of Systematic Reviews as a complementary source of meta-analyses. References of retrieved articles were then reviewed to identify additional studies. Following PRISMA [22], 2 researchers (HC and YQ) independently searched and screened related literature. The titles, abstracts, keywords, and full text of each study were reviewed for inclusion, and any ambiguity was resolved through discussion. Articles were included if they met the following inclusion criteria: (1) meta-analyses including prospective observational studies or studies that declared outcome-free at baseline; (2) evaluating the factors of any category associated with prostate cancer incidence; and (3) providing effect estimates for further calculation. The exclusion criteria were as follows: (1) meta-analyses including only retrospective studies; (2) narrative reviews or reviews without data synthesis results or failing to provide sufficient data for calculation; and (3) the outcome of interest was the diagnosis, treatment, or prognosis. The inclusion criteria for MR studies were similar but relatively concise: evaluating the factors of any category associated with

prostate cancer incidence using mendelian randomized analysis methods and providing effect estimates.

## Overlapping and outdated meta-analyses

For the same exposure factor evaluated by more than one meta-analysis published in different years, we preferentially selected the most recent or updated one including the largest number of studies (cohorts or datasets) with the maximum of participants to represent the best available evidence. The overlapping and outdated meta-analyses which were published earlier and contained fewer cohorts or datasets were thus excluded compared with selected one. For MR studies, we also selected the one which represented the best available evidence so far, taking into consideration the publication year, data source of both exposure and outcome, sample size, the proportion of variance ($r^2$) explained by selected instrumental variables (IVs), and the study quality comprehensively. The selection details of meta-analyses and MR studies were presented in S1 and S2 Tables, respectively.

## Data extraction and synthesis

A statistical analysis protocol in detail for this umbrella review was provided (S2 Text). In brief, in each included meta-analysis, qualified individual studies (cohort, case-cohort, or nested case-control study where exposure precedes the outcome) were selected, and relevant information were collected based on a predefined template: first author, publication year, study design, number of studies included, number of cases/population, ethnicity, exposure factors, outcomes of prostate cancer, comparisons, and effect estimates of any type, i.e., maximally adjusted hazard ratio (HR)/incidence rate ratio (IRR)/odds ratio (OR)/risk ratio (RR) with 95% confidence intervals, i.e., lower confidence interval (LCI) and upper confidence interval (UCI). Data extraction was conducted by 2 researchers (HC and YQ) separately and cross-check was performed to ensure correctness. Then, we repeated each meta-analysis based on extracted effect estimates, LCI, and UCI using the random-effects inverse-variance model with DerSimonian—Laird method. Heterogeneity between studies included in meta-analyses was represented using I square ($I^2$) value and Cochrane's Q P value [23]. $I^2 \leq 50\%$ was considered as no or small heterogeneity, and $I^2 > 50\%$ large heterogeneity. Publication bias was evaluated by using the Egger regression asymmetry test (significance threshold, $P < 0.10$) [24]. If the Egger's $P$ value was less than 0.1, we assumed the existence of publication bias. The 95% prediction interval (PI) estimated the middle 95% area of the predictive distribution and showed the range of true effects in future studies [25], reflecting the variation in the true effects across study settings. All statistical analyses were conducted with the use of Stata, version 14.0 (StataCorp), and R, version 3.3.0 (R Foundation for Statistical Computing).

From MR studies, we extracted key information of exposure, outcome, sample size, number of IVs, the variance ($r^2$) explained by IV, F statistics, and maximally adjusted effect estimates with 95% CI using the main analysis method, and no further calculation was needed for MR studies in this umbrella review.

## Quality assessment for included studies

The online 16-item AMSTAR-2 (A MeaSurement Tool to Assess systematic Reviews) checklist was used to assess methodological quality [26]. AMSTAR-2 considers the quality of the search, study inclusion and exclusion, description of individual studies, assessment of publication bias, heterogeneity, use of appropriate statistical methods, assessment of risk of bias in individual studies, and reporting of sources of funding and conflicts of interest. The items were scored

as No (0 point), Partial yes (0.5 point), or Yes (1 point). Both the total scores and critical item scores were calculated in our umbrella review [27].

For MR studies, quality assessment was performed with reference to the recently published STROBE-MR Statement (Strengthening the Reporting of Observational Studies in Epidemiology Using Mendelian Randomization) [28]. Briefly, the STROBE-MR checklist consists of 20 items that are grouped into sections Title and Abstract (item 1), Introduction (items 2 to 3), Methods (items 4 to 9), Results (items 10 to 13), Discussion (items 14 to 17), and Other Information (items 18 to 20). The checklist details were described elsewhere [28]. STROBE-MR puts emphasis on the transparent reporting of model assumptions assessment and sensitivity analyses, which also stands as a primary evaluation criterion in our review. Mendelian randomization assumptions regarding the reliability of IV (assumption 1) and absence of pleiotropic effects (assumption 2) were evaluated.

Two researchers (HC and ZT) rated the methodological quality of meta-analyses and reporting quality of MR studies and evaluated the assumptions of MR studies. In the case of disagreements, a decision was reached by consulting a third investigator (WZ).

## Evidence grading criteria for associations from meta-analyses

As shown in Table 1, the evidence credibility of statistically significant associations with prostate cancer was graded into 4 levels (convincing, highly suggestive, suggestive, and weak) based on precision of statistical significance, sample size, 95% PI, heterogeneity, and publication bias, with references to existing umbrella reviews [7,11,29]. Specifically, convincing evidence, as the highest level with the most stringent threshold, required summary estimate $P$ value <0.000001, large sample size (number of prostate cancer patients >1,000), no or small heterogeneity ($I^2 \leq 50\%$), no publication bias (Egger's $P \geq 0.10$), the largest component study (i.e., with the largest weight in meta-analysis) reporting directionally consistent with the overall estimate statistically significant association, and 95% PI excluding the null. Highly

**Table 1. Credibility assessment criteria for significant associations derived from meta-analyses of prospective observational studies and MR studies.**

| Evidence grading for meta-analyses | Detailed description |
|---|---|
| Convincing (I) | Significant associations with $P < 0.000001$; number of cases >1,000; the study with the largest weight reporting nominally significant results in the same direction as the overall estimate; 95% prediction interval excluding the null; no or small heterogeneity ($I^2 \leq 50\%$); no evidence of publication bias (Egger's $P$ value $\geq 0.10$). |
| Highly suggestive (II) | Associations with $P < 0.001$; number of cases >1,000; no or small heterogeneity ($I^2 \leq 50\%$); no evidence of publication bias (Egger's $P$ value $\geq 0.10$). |
| Suggestive (III) | Associations with $P < 0.05$; number of cases >1,000; the presence of large heterogeneity ($I^2 > 50\%$) or evidence of publication bias (Egger's $P$ value $< 0.10$). |
| Weak (IV) | Associations with $P < 0.05$; number of cases <1,000; the presence of large heterogeneity ($I^2 > 50\%$) and evidence of publication bias (Egger's $P$ value $< 0.10$). |
| **Evidence grading for MR studies** | **Detailed description** |
| Robust (I) | Significant associations with $P < 0.05$ across all analysis methods with consistent direction. |
| Probable (II) | Significant associations with $P < 0.05$ in at least 1 analysis method with consistent direction. |
| Suggestive (III) | Significant associations with $P < 0.05$ in at least 1 analysis method with inconsistent directions. |
| Insufficient (IV) | Significant associations with $P < 0.05$ based on 1 single analysis method (without sensitivity analysis). |

suggestive evidence, with the largest component study requirement removed, required a loosened effect P value threshold of <0.001, large sample size (number of prostate cancer patients >1,000), no or small heterogeneity ($I^2 \leq 50\%$), no evidence of publication bias (Egger's $P \geq 0.10$), and 95% PI excluding the null. Suggestive evidence required only statistical significance (P < 0.05), large sample size (number of prostate cancer patients >1,000), and allowed for the existence of either large heterogeneity ($I^2 > 50\%$) or publication bias (Egger's $P < 0.10$). Lastly, if one association was reported based on a case number less than 1,000, it would be defined as weak evidence due to insufficient statistical power. Also, associations showing the presence of both large heterogeneity and publication bias ($I^2 > 50\%$ and Egger's $P < 0.10$) would be graded as weak. Null associations were not included for evidence evaluation in this present umbrella review.

### Evidence grading criteria for causal associations from MR studies

We adopted and modified the evidence grading criteria categorized into robust, probable, suggestive, and insufficient proposed in the recently published MR review by Markozannes and colleagues [19]. The modified criteria excluded null associations and redefined the level of "insufficient" evidence. Briefly, robust evidence for causality was assigned based on nominally significant P value and directional concordant effect across all methods performed; probable evidence was assigned based on nominally significant P value in at least 1 method (main or sensitivity analyses) and concordant effect direction among all methods performed; suggestive evidence was assigned when at least 1 method had a nominally significant P value but the direction of the effect estimates differed between methods; insufficient evidence was assigned for significant associations based only on 1 main analysis while no sensitivity analysis was available (Table 1).

## Results

### Characteristics of included meta-analyses and summary on evidence grading

The process of literature identification and selection as well as updated work was recorded in detail in Fig 1. The initial search on January 10, 2022 yielded a total of 6,349 articles, and approximately 360 meta-analyses containing overlapped ones reporting on the same exposure published in different years were identified after excluding unrelated or duplicated articles. Then, 72 meta-analyses were selected for initial data synthesis. Updated search was conducted on September 9, 2023 upon request, yielding 1,015 newly published literature after the initial search, and 51 articles were included after excluding unrelated or duplicated articles. Then, 25 meta-analyses were selected for updated data synthesis, 5 of which replaced the previous ones. The selection of included meta-analyses was shown in S1 Table. Finally, in total 92 meta-analyses reporting 123 observational associations (Fig 2A) were included, categorized into 6 major categories: lifestyle [3,4,30–40] (N = 17); diet and nutrition [41–69] (N = 44); anthropometric indices [70–74] (N = 5); biomarkers [48,61,75–80] (N = 12); clinical variables, diseases, and treatments [81–112] (N = 39); and environmental factors [38,113–117] (N = 6). Note that the total number of associations was 123 while there were totally 122 factors because both the inverse association of finasteride with total prostate cancer and the positive association of finasteride with advanced prostate cancer were counted as 2 distinct associations.

The median (interquartile range, IQR) of studies (datasets) included in meta-analyses was 7 (4.25, 13), ranging from 2 to 35. The median (IQR) of case numbers in meta-analysis was

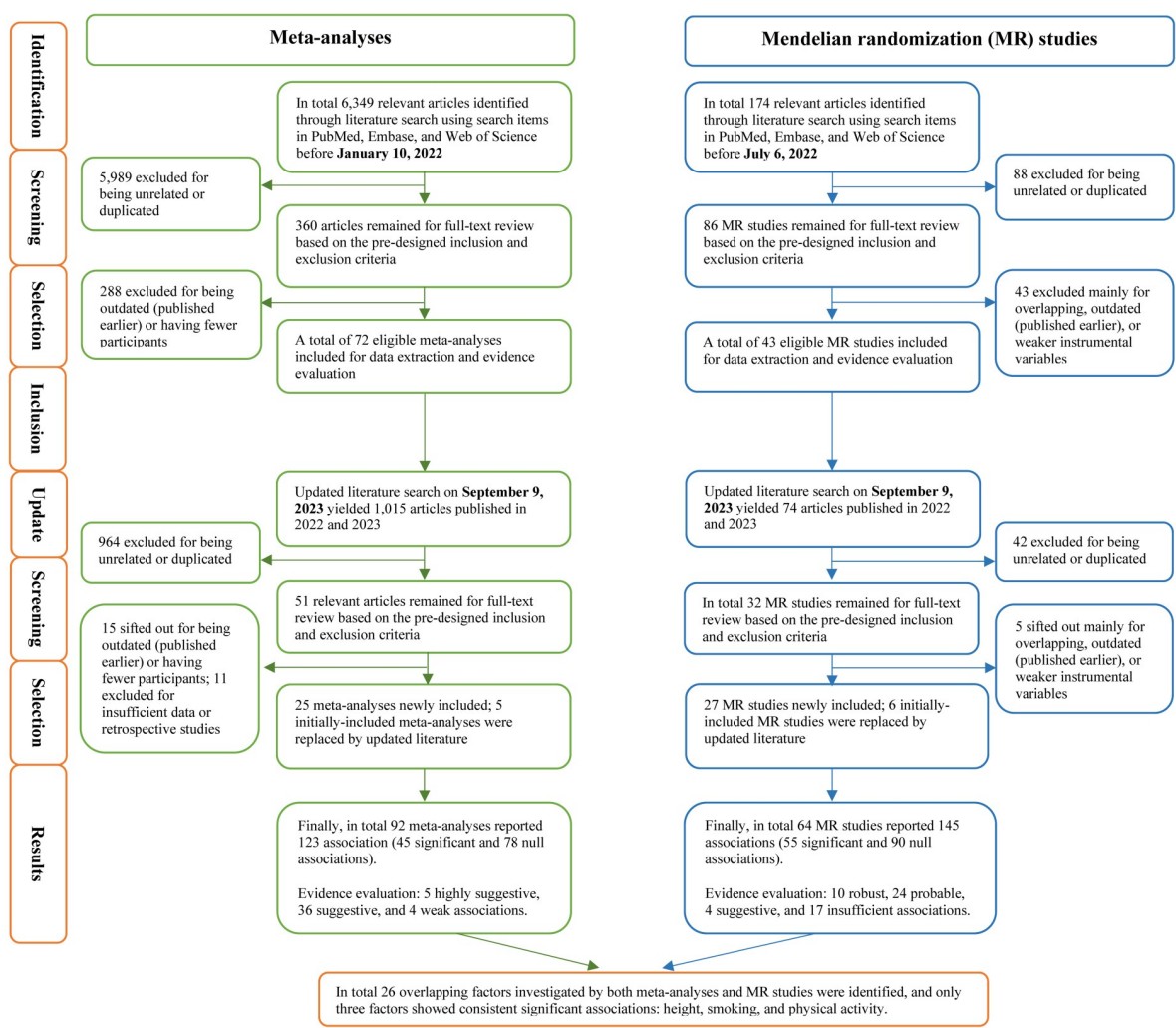

**Fig 1. Flowchart of literature search, inclusion, and results.** MR, mendelian randomization.

5,653 (2,735, 15,254), ranged from 20 to 118,077. The study design contained mostly cohort studies ($N = 1,342$, 95.7% of 1,403), with a small portion of nested case-control studies ($N = 50$, 3.6% of 1,403), case-cohort studies ($N = 4$, 0.2% of 1,403), and randomized controlled trials ($N = 7$, 0.5% of 1,403).

In total 90 eligible meta-analyses were assessed using AMSTAR-2 tool. The median (IQR) of AMSTAR-2 total score was 13.5 (13, 14) points, and that for AMSTAR-2 critical item score was 6 (5.5, 6) points. For the 7 AMSTAR-2 critical domains, 29% (26/90) of the included meta-analyses established a priori a protocol for the review, 100% (90/90) performed a comprehensive literature search, 71% (64/90) provided a list of excluded studies with justification, 93% (84/90) used a satisfactory technique for assessing the risk of bias in individual studies, 100% (90/90) used the appropriate model for meta-analysis, 74% (67/90) discussed the impact of risk of bias in individual studies in the interpretation of the results of the review, and 87% (78/90) performed graphical or statistical tests for publication bias and discussed the likelihood and magnitude of impact of publication bias (Fig 3). Each AMSTAR-2 domain judgment for each outcome is available in S3 Table.

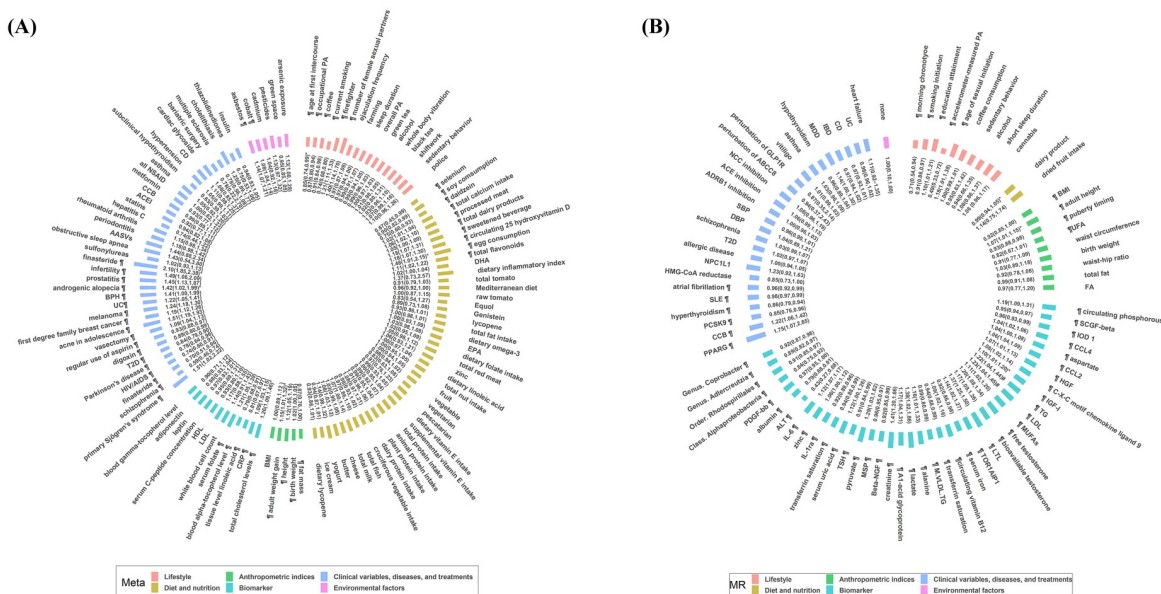

**Fig 2. Overall presentation of associations with the risk of prostate cancer.** (A) Observational associations from meta-analyses (Meta). (B) Causal associations from MR studies. Numbers presented in the graphs are OR with 95% confidence intervals. Different colors indicate different categories; ¶ represents significant associations (*P* < 0.05). Metrics with * denoting the outcome was advanced, aggressive, high-grade, or lethal prostate cancer, and metrics with # denoting the outcome was non-advanced, non-aggressive, or localized prostate cancer in graph (A). Metrics with * denoting the outcome of MR studies was aggressive prostate cancer, and metrics with # denoting the outcome of MR studies was early-onset prostate cancer in graph (B). Note that the null associations of biomarkers (*N* = 58) in MR studies are not presented here considering the graph size. Abbreviations in meta-analyses: PA, physical activity; DHA, docosahexaenoic acids; EPA, eicosapentaenoic; HDL, high-density lipoprotein; LDL, low-density lipoprotein; CRP, C-reactive protein; T2D, type 2 diabetes; BPH, benign prostate hyperplasia; HIV, human immunodeficiency virus; AIDS, acquired immune deficiency syndrome; CD, Crohn's disease; UC, ulcerative colitis; AASVs, anti-neutrophil cytoplasm antibody associated vasculitides; ACEI, angiotensin converting enzyme inhibitors; NSAID, nonsteroidal anti-inflammatory drug; CCB, calcium channel blockers. Abbreviations in MR studies: PA, physical activity; BMI, body mass index; UFA, unfavorable adiposity; FA, favorable adiposity; LTL, leukocyte telomere length; CCL2, Chemokine (C-C motif) ligand 2; CCL4, Chemokine (C-C motif) ligand 4; TG, triglyceride; IGF, insulin-like growth factor; LDL, low-density lipoprotein; HGF, hepatocyte growth factor; IL-1ra, IL-1 receptor antagonist; MUFAs, monounsaturated fatty acids; TOR1AIP1, Torsin-1A-interacting protein 1; IL-6, interleukin-6; ALT, alanine aminotransferase; IDO 1, Indoleamine 2,3-dioxygenase 1; PDGF-bb, platelet-derived growth factor BB; SCGF-β, stem cell growth factor-beta; TSH, thyroid-stimulating hormone; β-NGF, beta nerve growth factor; M.VLDL.TG, Triglycerides in medium VLDL; MSP, microseminoprotein-beta; CCB, calcium channel blockers; PCSK9, proprotein convertase subtilisin/kexin type 9; PPARG, peroxisome proliferator activated receptor γ; ABCC8, ATP binding cassette subfamily C member 8; GLP1R, glucagon-like peptide 1 receptor; ACE, angiotensin-converting enzyme; ADRB1, β-1 adrenergic receptor; NCC, sodium-chloride symporter; SBP, systolic blood pressure; DBP, diastolic blood pressure; MDD, major depressive disorder; SLE, systemic lupus erythematosus; IBD, inflammatory bowel disease; CD, Crohn's disease; UC, ulcerative colitis; T2D, type 2 diabetes; HMG-CoA, 3-hydroxy-3-methylglutaryl coenzyme A; NPC1L1, Niemann-Pick C1-Like 1. MR, mendelian randomization; OR, odds ratio.

In total 45 (of 123) significant associations (S1 Fig) were derived from 43 meta-analyses and subsequently graded, and the evidence grading details were elaborated in S4 Table. Among them, *P* values for summary effects were mostly between 0.001 and 0.05 (*N* = 27, 60% of 45) and between 0.001 and 0.000001 (*N* = 12, 27% of 45), while only 6 associations (*N* = 6, 13% of 45) had *P* values less than 0.000001. Only 3 associations (*N* = 3, 6.7% of 45) had case number of less than 1,000. Eleven associations (*N* = 11, 24% of 45) had 95% PI excluding the null. Twenty-three (*N* = 23, 51% of 45) associations reported presence of large heterogeneity (I² > 50%) and 9 (*N* = 9, 20% of 45) showed significant publication bias. In summary, there were 5 highly suggestive, 36 suggestive, and 4 weak associations in meta-analyses (Fig 4 and S5 Table). The remaining 78 associations were null and not graded.

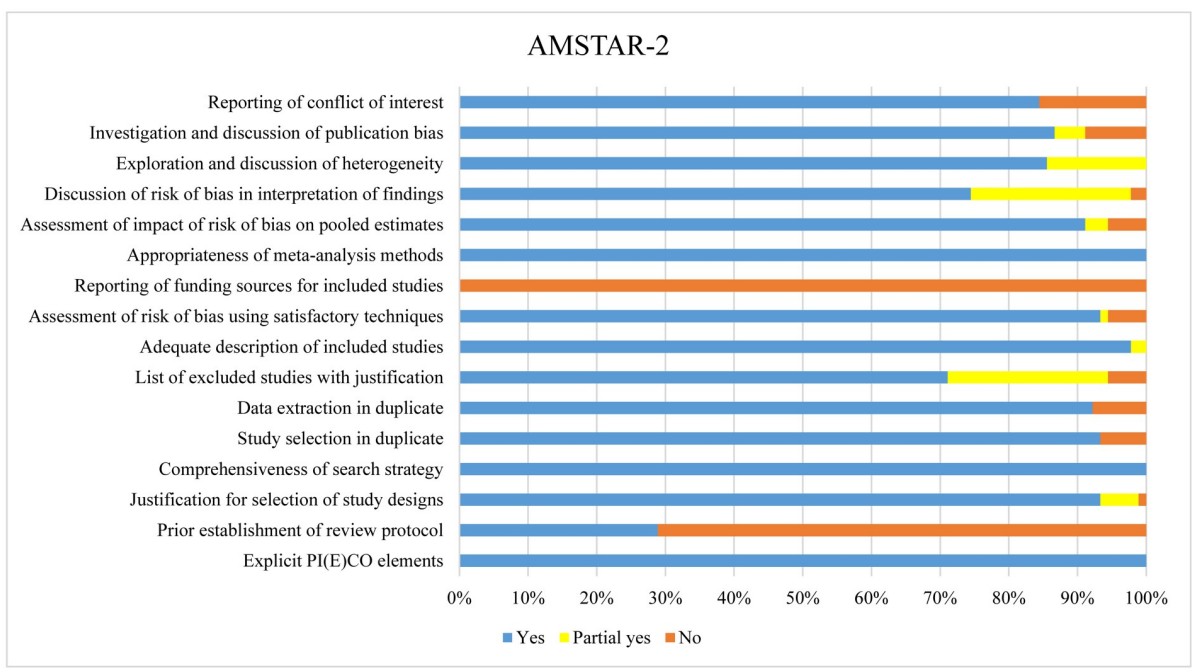

**Fig 3. Quality assessment of meta-analyses using AMSTAR-2.** The total number of meta-analyses included was 90. The items were scored as No (0 point), Partial yes (0.5 point), or Yes (1 point). Abbreviation: PI(E)CO, population, intervention or exposure, comparator, outcome.

Additionally, subgroup analyses of whites and non-whites population were performed for 11 significant associations from 11 meta-analyses (S6 Table and S2 Fig). As shown in S6 Table, the datasets, i.e., individual studies, in non-white populations were very limited compared to those in white populations. Five of the factors (firefighter, calcium, dairy products, height, and aspirin) were assessed in only 1 dataset in the corresponding meta-analysis [40,45,46,72,118] and 4 of the factors in 2 datasets [35,49,57,100]. There were 3 datasets available for ulcerative colitis (UC) [89] and 4 datasets for current smoking [30] in non-white populations. The subgroup analyses results showed the significant effects remained largely consistent for white population, while in non-white population, only the inverse associations of smoking and finasteride remained significant. In addition, total dairy products showed stronger effects in non-white population, though supported by only 1 study [119].

## Results of meta-analyses in categories

In total 17 lifestyle factors (of 123 total associations) were identified, of which 6 were significantly associated with prostate cancer (Fig 4 and S5 Table). Except for occupational physical activity reducing prostate cancer risk (OR = 0.87, 95% CI: 0.80, 0.94) as highly suggestive evidence, the remaining significant associations, including smoking (current smoking versus non-smoker, OR = 0.74, 95% CI: 0.68, 0.80), coffee (highest versus lowest, OR = 0.91, 95% CI: 0.84, 0.98), number of female sexual partners (highest versus lowest, OR = 1.40, 95% CI: 1.14, 1.70), age at first intercourse (highest versus lowest, OR = 0.85, 95% CI: 0.74, 0.99, for high-grade prostate cancer), and firefighter (ever-employment as a career firefighter versus general population, OR = 1.21, 95% CI: 1.11, 1.33) were all graded as suggestive. Null associations were found between prostate cancer and the following lifestyle factors: sleep duration (long or short), sedentary behavior, overall physical activity, green tea, black tea, alcohol, ejaculation frequency, shiftwork, whole body vibration, farming, and police.

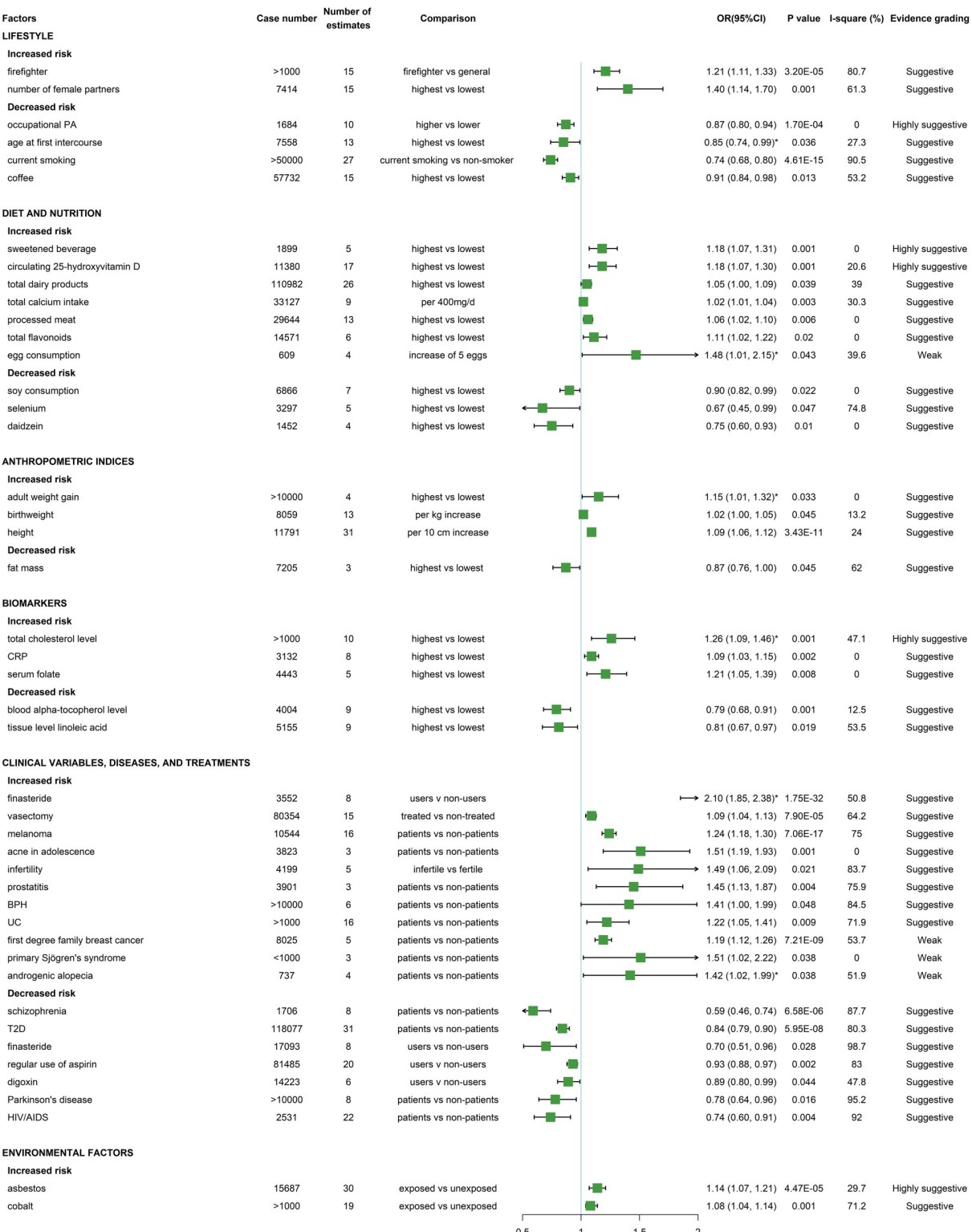

| Factors | Case number | Number of estimates | Comparison | OR(95%CI) | P value | I-square (%) | Evidence grading |
|---|---|---|---|---|---|---|---|
| **LIFESTYLE** | | | | | | | |
| **Increased risk** | | | | | | | |
| firefighter | >1000 | 15 | firefighter vs general | 1.21 (1.11, 1.33) | 3.20E-05 | 80.7 | Suggestive |
| number of female partners | 7414 | 15 | highest vs lowest | 1.40 (1.14, 1.70) | 0.001 | 61.3 | Suggestive |
| **Decreased risk** | | | | | | | |
| occupational PA | 1684 | 10 | higher vs lower | 0.87 (0.80, 0.94) | 1.70E-04 | 0 | Highly suggestive |
| age at first intercourse | 7558 | 13 | highest vs lowest | 0.85 (0.74, 0.99)* | 0.036 | 27.3 | Suggestive |
| current smoking | >50000 | 27 | current smoking vs non-smoker | 0.74 (0.68, 0.80) | 4.61E-15 | 90.5 | Suggestive |
| coffee | 57732 | 15 | highest vs lowest | 0.91 (0.84, 0.98) | 0.013 | 53.2 | Suggestive |
| **DIET AND NUTRITION** | | | | | | | |
| **Increased risk** | | | | | | | |
| sweetened beverage | 1899 | 5 | highest vs lowest | 1.18 (1.07, 1.31) | 0.001 | 0 | Highly suggestive |
| circulating 25-hydroxyvitamin D | 11380 | 17 | highest vs lowest | 1.18 (1.07, 1.30) | 0.001 | 20.6 | Highly suggestive |
| total dairy products | 110982 | 26 | highest vs lowest | 1.05 (1.00, 1.09) | 0.039 | 39 | Suggestive |
| total calcium intake | 33127 | 9 | per 400mg/d | 1.02 (1.01, 1.04) | 0.003 | 30.3 | Suggestive |
| processed meat | 29644 | 13 | highest vs lowest | 1.06 (1.02, 1.10) | 0.006 | 0 | Suggestive |
| total flavonoids | 14571 | 6 | highest vs lowest | 1.11 (1.02, 1.22) | 0.02 | 0 | Suggestive |
| egg consumption | 609 | 4 | increase of 5 eggs | 1.48 (1.01, 2.15)* | 0.043 | 39.6 | Weak |
| **Decreased risk** | | | | | | | |
| soy consumption | 6866 | 7 | highest vs lowest | 0.90 (0.82, 0.99) | 0.022 | 0 | Suggestive |
| selenium | 3297 | 5 | highest vs lowest | 0.67 (0.45, 0.99) | 0.047 | 74.8 | Suggestive |
| daidzein | 1452 | 4 | highest vs lowest | 0.75 (0.60, 0.93) | 0.01 | 0 | Suggestive |
| **ANTHROPOMETRIC INDICES** | | | | | | | |
| **Increased risk** | | | | | | | |
| adult weight gain | >10000 | 4 | highest vs lowest | 1.15 (1.01, 1.32)* | 0.033 | 0 | Suggestive |
| birthweight | 8059 | 13 | per kg increase | 1.02 (1.00, 1.05) | 0.045 | 13.2 | Suggestive |
| height | 11791 | 31 | per 10 cm increase | 1.09 (1.06, 1.12) | 3.43E-11 | 24 | Suggestive |
| **Decreased risk** | | | | | | | |
| fat mass | 7205 | 3 | highest vs lowest | 0.87 (0.76, 1.00) | 0.045 | 62 | Suggestive |
| **BIOMARKERS** | | | | | | | |
| **Increased risk** | | | | | | | |
| total cholesterol level | >1000 | 10 | highest vs lowest | 1.26 (1.09, 1.46)* | 0.001 | 47.1 | Highly suggestive |
| CRP | 3132 | 8 | highest vs lowest | 1.09 (1.03, 1.15) | 0.002 | 0 | Suggestive |
| serum folate | 4443 | 5 | highest vs lowest | 1.21 (1.05, 1.39) | 0.008 | 0 | Suggestive |
| **Decreased risk** | | | | | | | |
| blood alpha-tocopherol level | 4004 | 9 | highest vs lowest | 0.79 (0.68, 0.91) | 0.001 | 12.5 | Suggestive |
| tissue level linoleic acid | 5155 | 9 | highest vs lowest | 0.81 (0.67, 0.97) | 0.019 | 53.5 | Suggestive |
| **CLINICAL VARIABLES, DISEASES, AND TREATMENTS** | | | | | | | |
| **Increased risk** | | | | | | | |
| finasteride | 3552 | 8 | users v non-users | 2.10 (1.85, 2.38)* | 1.75E-32 | 50.8 | Suggestive |
| vasectomy | 80354 | 15 | treated vs non-treated | 1.09 (1.04, 1.13) | 7.90E-05 | 64.2 | Suggestive |
| melanoma | 10544 | 16 | patients vs non-patients | 1.24 (1.18, 1.30) | 7.06E-17 | 75 | Suggestive |
| acne in adolescence | 3823 | 3 | patients vs non-patients | 1.51 (1.19, 1.93) | 0.001 | 0 | Suggestive |
| infertility | 4199 | 5 | infertile vs fertile | 1.49 (1.06, 2.09) | 0.021 | 83.7 | Suggestive |
| prostatitis | 3901 | 3 | patients vs non-patients | 1.45 (1.13, 1.87) | 0.004 | 75.9 | Suggestive |
| BPH | >10000 | 6 | patients vs non-patients | 1.41 (1.00, 1.99) | 0.048 | 84.5 | Suggestive |
| UC | >1000 | 16 | patients vs non-patients | 1.22 (1.05, 1.41) | 0.009 | 71.9 | Suggestive |
| first degree family breast cancer | 8025 | 5 | patients vs non-patients | 1.19 (1.12, 1.26) | 7.21E-09 | 53.7 | Weak |
| primary Sjögren's syndrome | <1000 | 3 | patients vs non-patients | 1.51 (1.02, 2.22) | 0.038 | 0 | Weak |
| androgenic alopecia | 737 | 4 | patients vs non-patients | 1.42 (1.02, 1.99)* | 0.038 | 51.9 | Weak |
| **Decreased risk** | | | | | | | |
| schizophrenia | 1706 | 8 | patients vs non-patients | 0.59 (0.46, 0.74) | 6.58E-06 | 87.7 | Suggestive |
| T2D | 118077 | 31 | patients vs non-patients | 0.84 (0.79, 0.90) | 5.95E-08 | 80.3 | Suggestive |
| finasteride | 17093 | 8 | users vs non-users | 0.70 (0.51, 0.96) | 0.028 | 98.7 | Suggestive |
| regular use of aspirin | 81485 | 20 | users v non-users | 0.93 (0.88, 0.97) | 0.002 | 83 | Suggestive |
| digoxin | 14223 | 6 | users v non-users | 0.89 (0.80, 0.99) | 0.044 | 47.8 | Suggestive |
| Parkinson's disease | >10000 | 8 | patients vs non-patients | 0.78 (0.64, 0.96) | 0.016 | 95.2 | Suggestive |
| HIV/AIDS | 2531 | 22 | patients vs non-patients | 0.74 (0.60, 0.91) | 0.004 | 92 | Suggestive |
| **ENVIRONMENTAL FACTORS** | | | | | | | |
| **Increased risk** | | | | | | | |
| asbestos | 15687 | 30 | exposed vs unexposed | 1.14 (1.07, 1.21) | 4.47E-05 | 29.7 | Highly suggestive |
| cobalt | >1000 | 19 | exposed vs unexposed | 1.08 (1.04, 1.14) | 0.001 | 71.2 | Suggestive |

**Fig 4. Forest plot of evidence grading for significant associations with the risk of prostate cancer in categories from meta-analyses.** The statistical test to determine the *P* value in meta-analyses was the random-effects inverse-variance model with DerSimonian—Laird method. The pooled effect estimate OR of each association is represented by the green colored square and 95% CI by the horizontal lines. Metrics with * denoting the outcome was high-grade, aggressive, or advanced prostate cancer. PA, physical activity; CRP, C-reactive protein; T2D, type 2 diabetes; BPH, benign prostate hyperplasia; UC, ulcerative colitis; HIV, human immunodeficiency virus; AIDS, acquired immune deficiency syndrome; OR, odds ratio.

A total of 44 diet and nutritional factors (of 123 total associations) were included in this review, and 10 of them showed significant associations with prostate cancer (Fig 4 and S5 Table). Highly suggestive evidence was observed for sweetened beverage (highest versus lowest, OR = 1.18, 95% CI: 1.07, 1.31) and circulating 25-hydroxyvitamin D (high versus low, OR = 1.18, 95% CI: 1.07, 1.30). Suggestive evidence was observed for daidzein (highest versus lowest, OR = 0.75, 95% CI: 0.60, 0.93), selenium (highest versus lowest, OR = 0.67, 95% CI: 0.45, 0.99), total flavonoids (highest versus lowest, OR = 1.11, 95% CI: 1.02, 1.22), and 4 other factors with only marginal effect including total dairy products (highest versus lowest, OR = 1.05, 95% CI: 1.00, 1.09), processed meat (highest versus lowest, OR = 1.06, 95% CI: 1.02, 1.10), total calcium intake (per 400 mg/d, RR = 1.02, 95% CI: 1.01, 1.04), and soy consumption (highest versus lowest, OR = 0.90, 95% CI: 0.82, 0.99). Egg consumption (increase of 5 eggs, OR = 1.48, 95% CI: 1.01, 2.15) increasing high-grade prostate cancer risk was graded as weak evidence mainly due to the small number of patients (less than 1,000). Null associations with prostate cancer (N = 34) were found for docosahexaenoic acids (DHAs), eicosapentaenoic (EPA), dietary omega-3, genistein, equol, dietary lycopene, dietary phosphorus intake, dietary linoleic acid, dietary inflammatory index, Mediterranean diet, dietary folate intake, dietary vitamin E intake, supplemental vitamin E intake, total protein intake, animal protein intake, plant protein intake, dairy protein intake, cruciferous vegetable intake, total fish, zinc, raw tomato, total tomato, total nut intake, fruit, vegetable, vegetarian, pescatarian, red meat, cheese, butter, yogurt, ice cream, dietary folate intake, and total fat intake.

Five anthropometric indices (of 123 total associations) were included (Fig 4 and S5 Table) and 4 including birth weight (per kg increase, OR = 1.02, 95% CI: 1.00, 1.05), height (per 10 cm increase, OR = 1.09, 95% CI: 1.06, 1.12), and fat mass (highest versus lowest, OR = 0.87, 95% CI: 0.76, 1.00) were significantly associated with total prostate cancer risk and adult weight gain with high-risk prostate cancer (highest versus lowest, OR = 1.15, 95% CI: 1.01, 1.32), all with small effect and graded as suggestive evidence. Body mass index (BMI) was not found to associate with prostate cancer according to the selected meta-analysis [70].

In total 12 biomarkers (of 123 total associations) were included, with 5 showing significant association with prostate cancer (Fig 4 and S5 Table). Total cholesterol level was associated with increased risk of high-grade prostate cancer (highest versus lowest, OR = 1.26, 95% CI: 1.09, 1.46), which was highly suggestive. C-reactive protein (CRP) (highest versus lowest quartiles, OR = 1.09, 95% CI: 1.03, 1.15), serum folate (highest versus lowest, OR = 1.21, 95% CI: 1.05, 1.39), tissue level linoleic acid (highest versus lowest, OR = 0.81, 95% CI: 0.67, 0.97), and blood α-tocopherol level (highest versus lowest, OR = 0.79, 95% CI: 0.68, 0.91) showed significant association and were all graded as suggestive. The rest of included biomarkers blood γ-tocopherol level, high-density lipoprotein (HDL), low-density lipoprotein (LDL), leptin, adiponectin, serum C-peptide concentration, and white blood cell count exhibited null association with prostate cancer.

Totally 39 clinical variables, diseases, and treatments (of 123 total associations) were included in this review, with almost half significantly associated with prostate cancer risk and mostly graded as suggestive evidence (Fig 4 and S5 Table). Among the 18 significant associations, 11 factors were associated with higher prostate cancer risk including melanoma (patients versus non-patients, OR = 1.24, 95% CI: 1.18, 1.30), acne in adolescence (patients versus non-patients, OR = 1.51, 95% CI: 1.19, 1.93), infertility (infertile versus fertile, OR = 1.49, 95% CI: 1.06, 2.09), prostatitis (patients versus non-patients, OR = 1.45, 95% CI: 1.13, 1.87), benign prostatic hyperplasia (BPH) (patients versus non-patients, OR = 1.41, 95% CI: 1.00, 1.99), vasectomy (treated versus non-treated, OR = 1.09, 95% CI: 1.04, 1.13), and finasteride with high-grade prostate cancer (users versus non-users, OR = 2.10, 95% CI: 1.85, 2.38), graded as suggestive evidence, and first-degree family breast cancer (patients versus non-patients,

OR = 1.19, 95% CI: 1.12, 1.26), UC (patients versus non-patients, OR = 1.22, 95% CI: 1.05, 1.41), primary Sjögren's syndrome (patients versus non-patients, OR = 1.51, 95% CI: 1.02, 2.22), and androgenic alopecia for high-grade prostate cancer (patients versus non-patients, OR = 1.42, 95% CI: 1.02, 1.99) as weak evidence. In addition, 7 clinical variables, diseases, and treatments were inversely associated with prostate cancer risk, including type 2 diabetes (T2D) (patients versus non-patients, OR = 0.84, 95% CI: 0.79, 0.90), Parkinson's disease (patients versus non-patients, OR = 0.78, 95% CI: 0.64, 0.96), schizophrenia (patients versus non-patients, OR = 0.59, 95% CI: 0.46, 0.74), regular use of aspirin (patients versus non-patients, OR = 0.93, 95% CI: 0.88, 0.97), digoxin (patients versus non-patients, OR = 0.89, 95% CI: 0.80, 0.99), and finasteride (users versus non-users, OR = 0.70, 95% CI: 0.51, 0.96) graded as suggestive evidence except HIV/AIDS (patients versus non-patients, OR = 0.74, 95% CI: 0.60, 0.91) as weak evidence. Interestingly, opposite associations found in finasteride, which decreased risk of total prostate cancer but increased risk of high-grade prostate cancer, both as suggestive evidence. The remaining clinical variables, diseases, and treatments showing no significant associations with prostate cancer were hepatitis C, periodontitis, asthma, Crohn's disease, rheumatoid arthritis, anti-neutrophil cytoplasm antibody associated vasculitides (AASVs), hypertension, obstructive sleep apnea, subclinical hypothyroidism, bariatric surgery, multiple sclerosis, cholelithiasis, metformin, statins, angiotensin converting enzyme inhibitors (ACEI), calcium-channel blockers (CCB), thiazolidinediones, sulfonylureas, insulin, cardiac glycoside, and nonsteroidal anti-inflammatory drug (NSAID).

Six environmental factors (of 123 total associations) were identified in this review, with 2 factors significantly associated with prostate cancer risk (Fig 4 and S5 Table). Asbestos (exposed versus unexposed, OR = 1.14, 95% CI: 1.07, 1.21) and cobalt (exposed versus unexposed, OR = 1.08, 95% CI: 1.04, 1.14) increasing the risk of prostate cancer were graded as highly suggestive and suggestive evidence, respectively. Cadmium, pesticides, green space, and arsenic exposure had no significant association with prostate cancer.

## Characteristics of included MR studies and summary on evidence grading results

As shown in Fig 1, the initial search on July 6, 2022 yielded a total of 174 articles, approximately 86 MR studies containing overlapped ones reporting on the same exposure published in different years were identified after excluding unrelated or duplicated articles, and then 43 were initially selected. Updated search on September 9, 2023 yielded 74 newly published literature, and 32 articles were included after excluding unrelated or duplicated articles. Then, 27 were selected for updated data synthesis, 6 of which replaced the previous ones. The selection of included MR studies was shown in S2 Table. Finally, 64 MR studies investigated 145 associations (Fig 2B) categorized into lifestyle [16,120–127] (N = 10); diet and nutrition [125,128] (N = 2); anthropometric indices [125,129,130] (N = 9); biomarkers [17,60,125,131–163] (N = 98); clinical variables, diseases, and treatments [93,163–176] (N = 26), and environmental factors (N = 0) (S7 Table). Particularly, over 200 biomarkers including amino acids and derivative, fatty acids and derivatives, growth factors, inflammatory biomarkers, lipid metabolism biomarkers, methylations, other metabolites/biomarkers, steroids, and circulating leukocyte telomere length were well documented in the previous review [19], and hence only significant associations (N = 18) were selected and discussed in this present review. All studies used two-sample MR design, with European ancestry outcome data mostly from PRACTICAL (The Prostate Cancer Association Group to Investigate Cancer Associated Alterations in the Genome consortium) (N = 113, 78% of 145 total associations). The median (IQR) of number of IVs was 13.5 (4, 54.25), ranging from 1 to 663. All studies were in line with the

STROBE-MR, demonstrating good reporting quality. Concerning sensitivity analysis, there were 94 associations (65% of 145 total associations) reporting sensitivity analysis results. In total 55 significant causal associations (of 145 total associations) from MR studies were graded. Finally, 10 causal associations were assigned robust, 24 probable, 4 suggestive, and 17 insufficient (Fig 5 and S7 Table).

## Results of MR studies in categories

Ten lifestyle factors (of 145 total associations) were included, with 5 showing significant causal associations (Fig 5 and S7 Table). Robust evidence was assigned to morning chronotype (1 h earlier, OR = 0.71, 95% CI: 0.54, 0.94) and smoking initiation (1 standard deviation (SD) increase, OR = 0.91, 95% CI: 0.86, 0.97). Probable evidence was assigned to education attainment (per SD increase in genetically predicted years of education, OR = 1.10, 95% CI: 1.01, 1.21). Suggestive evidence was assigned to age of sexual initiation (older age, OR = 1.18, 95% CI: 1.01, 1.38). Insufficient evidence was observed for accelerator-measured physical activity (per SD increase, OR = 0.49, 95% CI: 0.33, 0.72), causally reducing prostate cancer risk. Remaining lifestyle factors, namely coffee consumption, alcohol, cannabis, short sleep duration, and sedentary behavior demonstrated no causal relationship with prostate cancer.

Only 2 diet and nutrition factors (of 145 total associations) including dairy products (milk intake) and dried fruit intake were identified and both showed no evidence of causality (S7 Table).

In total 9 anthropometric indices (of 145 total associations) were identified, with 4 significant causally associated with prostate cancer (Fig 5 and S7 Table). Robust evidence was assigned to BMI (per SD, OR = 0.92, 95% CI: 0.85, 1.00), and insufficient evidence was assigned to height for high-grade prostate cancer (per SD, OR = 1.07, 95% CI: 1.01, 1.15) and puberty timing (later puberty, OR = 0.93, 95% CI: 0.88, 0.98). Probable evidence was assigned to unfavorable adiposity (UFA), which met the evidence criteria for probable though the *P* value for main analysis was larger than 0.05. While 5 factors including birth weight, waist circumference, waist-hip ratio, favorable adiposity (FA), and total fat showed null causality with prostate cancer.

A total of 98 biomarkers (of 145 total associations) were included, with 40 biomarkers significantly associated with prostate cancer (Fig 5 and S7 Table). Robust evidence was observed for circulating phosphorous (per SD, OR = 1.19, 95% CI: 1.09, 1.31), leukocyte telomere length (LTL) (long versus short, OR = 1.37, 95% CI: 1.25, 1.50), serum uric acid (per SD, OR = 1.12, 95% CI: 1.00, 1.26) increasing risk of prostate cancer, and alanine aminotransferase (per SD, OR = 0.43, 95% CI: 0.27, 0.68), albumin (per SD, OR = 0.79, 95% CI: 0.68, 0.91) reducing risk of prostate cancer. In addition, there were 19 probable (vitamin B12, transferrin saturation, alanine, Chemokine (C-C motif) ligand 2, Chemokine (C-C motif) ligand 4, C-X-C motif chemokine ligand 9, triglyceride, insulin-like growth factor 1, LDL, bioavailable testosterone, free testosterone, hepatocyte growth factor, IL-1 receptor antagonist, Indoleamine 2,3-dioxygenase 1, platelet-derived growth factor BB, stem cell growth factor-beta, Class. Alphaproteobacteria, Order. Rhodospirillales, and Genus. Adlercreutzia), 3 suggestive (monounsaturated fatty acids, aspartate, and Genus. Coprobacter), and 13 insufficient associations (a1-acid glycoprotein, Torsin-1A-interacting protein 1, pyruvate, lactate, creatinine, alanine, zinc, interleukin-6, serum iron, thyroid-stimulating hormone, beta nerve growth factor; triglycerides in medium VLDL, and microseminoprotein-beta). The remaining 58 biomarkers showed null association with prostate cancer.

Totally 26 clinical variables, diseases, and treatments (of 145 total associations) were included, with 6 showing significant causal association with prostate cancer (Fig 5 and S7

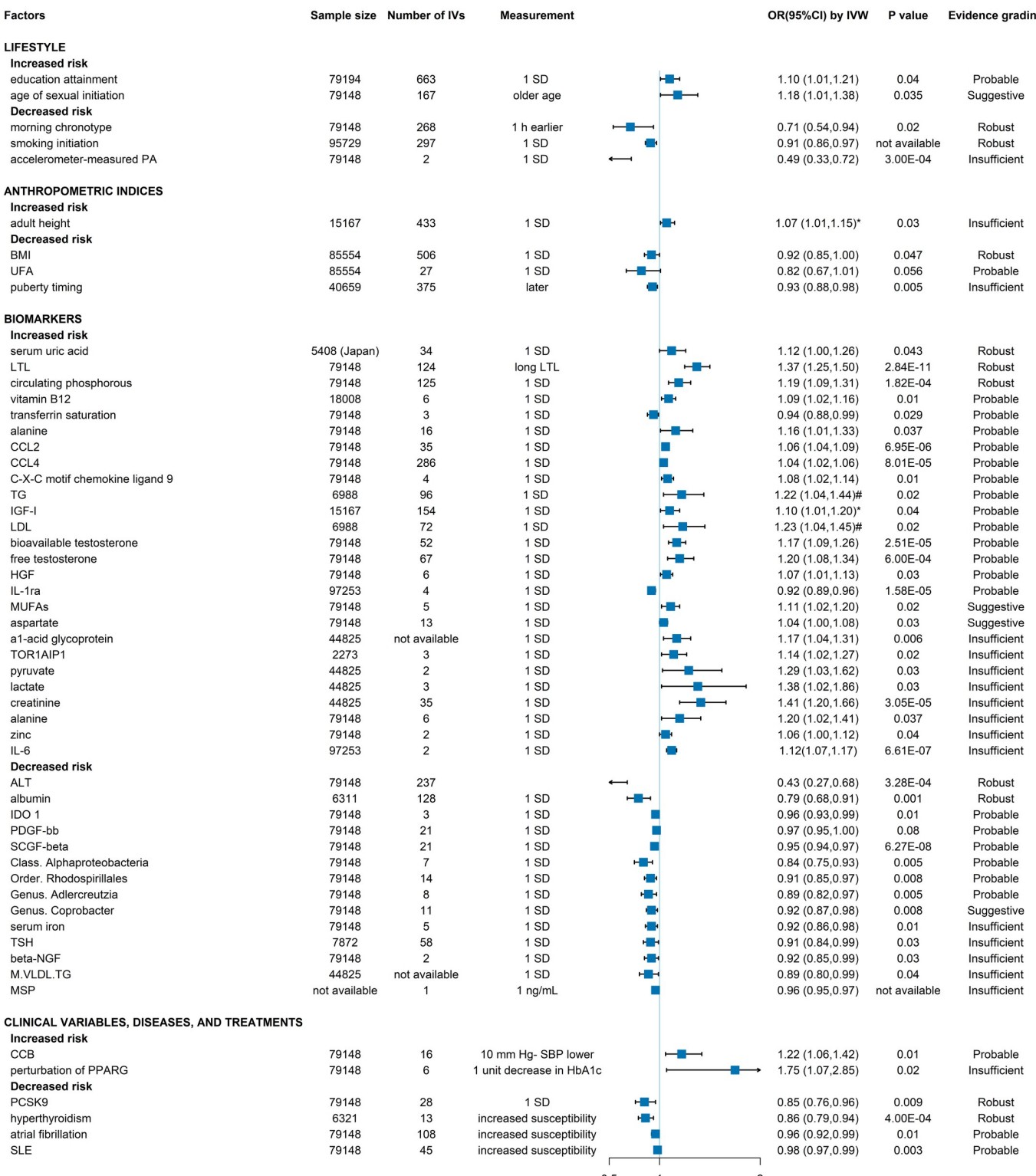

| Factors | Sample size | Number of IVs | Measurement | | OR(95%CI) by IVW | P value | Evidence grading |
|---|---|---|---|---|---|---|---|
| **LIFESTYLE** | | | | | | | |
| **Increased risk** | | | | | | | |
| education attainment | 79194 | 663 | 1 SD | | 1.10 (1.01,1.21) | 0.04 | Probable |
| age of sexual initiation | 79148 | 167 | older age | | 1.18 (1.01,1.38) | 0.035 | Suggestive |
| **Decreased risk** | | | | | | | |
| morning chronotype | 79148 | 268 | 1 h earlier | | 0.71 (0.54,0.94) | 0.02 | Robust |
| smoking initiation | 95729 | 297 | 1 SD | | 0.91 (0.86,0.97) | not available | Robust |
| accelerometer-measured PA | 79148 | 2 | 1 SD | | 0.49 (0.33,0.72) | 3.00E-04 | Insufficient |
| **ANTHROPOMETRIC INDICES** | | | | | | | |
| **Increased risk** | | | | | | | |
| adult height | 15167 | 433 | 1 SD | | 1.07 (1.01,1.15)* | 0.03 | Insufficient |
| **Decreased risk** | | | | | | | |
| BMI | 85554 | 506 | 1 SD | | 0.92 (0.85,1.00) | 0.047 | Robust |
| UFA | 85554 | 27 | 1 SD | | 0.82 (0.67,1.01) | 0.056 | Probable |
| puberty timing | 40659 | 375 | later | | 0.93 (0.88,0.98) | 0.005 | Insufficient |
| **BIOMARKERS** | | | | | | | |
| **Increased risk** | | | | | | | |
| serum uric acid | 5408 (Japan) | 34 | 1 SD | | 1.12 (1.00,1.26) | 0.043 | Robust |
| LTL | 79148 | 124 | long LTL | | 1.37 (1.25,1.50) | 2.84E-11 | Robust |
| circulating phosphorous | 79148 | 125 | 1 SD | | 1.19 (1.09,1.31) | 1.82E-04 | Robust |
| vitamin B12 | 18008 | 6 | 1 SD | | 1.09 (1.02,1.16) | 0.01 | Probable |
| transferrin saturation | 79148 | 3 | 1 SD | | 0.94 (0.88,0.99) | 0.029 | Probable |
| alanine | 79148 | 16 | 1 SD | | 1.16 (1.01,1.33) | 0.037 | Probable |
| CCL2 | 79148 | 35 | 1 SD | | 1.06 (1.04,1.09) | 6.95E-06 | Probable |
| CCL4 | 79148 | 286 | 1 SD | | 1.04 (1.02,1.06) | 8.01E-05 | Probable |
| C-X-C motif chemokine ligand 9 | 79148 | 4 | 1 SD | | 1.08 (1.02,1.14) | 0.01 | Probable |
| TG | 6988 | 96 | 1 SD | | 1.22 (1.04,1.44)# | 0.02 | Probable |
| IGF-I | 15167 | 154 | 1 SD | | 1.10 (1.01,1.20)* | 0.04 | Probable |
| LDL | 6988 | 72 | 1 SD | | 1.23 (1.04,1.45)# | 0.02 | Probable |
| bioavailable testosterone | 79148 | 52 | 1 SD | | 1.17 (1.09,1.26) | 2.51E-05 | Probable |
| free testosterone | 79148 | 67 | 1 SD | | 1.20 (1.08,1.34) | 6.00E-04 | Probable |
| HGF | 79148 | 6 | 1 SD | | 1.07 (1.01,1.13) | 0.03 | Probable |
| IL-1ra | 97253 | 4 | 1 SD | | 0.92 (0.89,0.96) | 1.58E-05 | Probable |
| MUFAs | 79148 | 5 | 1 SD | | 1.11 (1.02,1.20) | 0.02 | Suggestive |
| aspartate | 79148 | 13 | 1 SD | | 1.04 (1.00,1.08) | 0.03 | Suggestive |
| a1-acid glycoprotein | 44825 | not available | 1 SD | | 1.17 (1.04,1.31) | 0.006 | Insufficient |
| TOR1AIP1 | 2273 | 3 | 1 SD | | 1.14 (1.02,1.27) | 0.02 | Insufficient |
| pyruvate | 44825 | 2 | 1 SD | | 1.29 (1.03,1.62) | 0.03 | Insufficient |
| lactate | 44825 | 3 | 1 SD | | 1.38 (1.02,1.86) | 0.03 | Insufficient |
| creatinine | 44825 | 35 | 1 SD | | 1.41 (1.20,1.66) | 3.05E-05 | Insufficient |
| alanine | 79148 | 6 | 1 SD | | 1.20 (1.02,1.41) | 0.037 | Insufficient |
| zinc | 79148 | 2 | 1 SD | | 1.06 (1.00,1.12) | 0.04 | Insufficient |
| IL-6 | 97253 | 2 | 1 SD | | 1.12(1.07,1.17) | 6.61E-07 | Insufficient |
| **Decreased risk** | | | | | | | |
| ALT | 79148 | 237 | | | 0.43 (0.27,0.68) | 3.28E-04 | Robust |
| albumin | 6311 | 128 | 1 SD | | 0.79 (0.68,0.91) | 0.001 | Robust |
| IDO 1 | 79148 | 3 | 1 SD | | 0.96 (0.93,0.99) | 0.01 | Probable |
| PDGF-bb | 79148 | 21 | 1 SD | | 0.97 (0.95,1.00) | 0.08 | Probable |
| SCGF-beta | 79148 | 21 | 1 SD | | 0.95 (0.94,0.97) | 6.27E-08 | Probable |
| Class. Alphaproteobacteria | 79148 | 7 | 1 SD | | 0.84 (0.75,0.93) | 0.005 | Probable |
| Order. Rhodospirillales | 79148 | 14 | 1 SD | | 0.91 (0.85,0.97) | 0.008 | Probable |
| Genus. Adlercreutzia | 79148 | 8 | 1 SD | | 0.89 (0.82,0.97) | 0.005 | Probable |
| Genus. Coprobacter | 79148 | 11 | 1 SD | | 0.92 (0.87,0.98) | 0.008 | Suggestive |
| serum iron | 79148 | 5 | 1 SD | | 0.92 (0.86,0.98) | 0.01 | Insufficient |
| TSH | 7872 | 58 | 1 SD | | 0.91 (0.84,0.99) | 0.03 | Insufficient |
| beta-NGF | 79148 | 2 | 1 SD | | 0.92 (0.85,0.99) | 0.03 | Insufficient |
| M.VLDL.TG | 44825 | not available | 1 SD | | 0.89 (0.80,0.99) | 0.04 | Insufficient |
| MSP | not available | 1 | 1 ng/mL | | 0.96 (0.95,0.97) | not available | Insufficient |
| **CLINICAL VARIABLES, DISEASES, AND TREATMENTS** | | | | | | | |
| **Increased risk** | | | | | | | |
| CCB | 79148 | 16 | 10 mm Hg- SBP lower | | 1.22 (1.06,1.42) | 0.01 | Probable |
| perturbation of PPARG | 79148 | 6 | 1 unit decrease in HbA1c | | 1.75 (1.07,2.85) | 0.02 | Insufficient |
| **Decreased risk** | | | | | | | |
| PCSK9 | 79148 | 28 | 1 SD | | 0.85 (0.76,0.96) | 0.009 | Robust |
| hyperthyroidism | 6321 | 13 | increased susceptibility | | 0.86 (0.79,0.94) | 4.00E-04 | Robust |
| atrial fibrillation | 79148 | 108 | increased susceptibility | | 0.96 (0.92,0.99) | 0.01 | Probable |
| SLE | 79148 | 45 | increased susceptibility | | 0.98 (0.97,0.99) | 0.003 | Probable |

0.5   1   2

**Fig 5. Forest plot of evidence grading for significant associations with the risk of prostate cancer in categories from MR studies.** The statistical test to determine the *P* value in MR study was the IVW regression analysis. The effect estimate OR of each association is represented by the blue colored square and 95% CI by the horizontal lines. Metrics with * denoting the outcome was high-grade, aggressive, or advanced prostate cancer. Metrics with # denoting the outcome was early-onset prostate cancer. Note that UFA meets the evidence criteria for probable though the *P* value for main analysis is larger than 0.05. SD, standard deviation; PA, physical activity; BMI, body mass index; LTL, leukocyte telomere length; CCL2, Chemokine (C-C motif) ligand 2; CCL4, Chemokine (C-C motif)

ligand 4; TG, triglyceride; IGF, insulin-like growth factor; LDL, low-density lipoprotein; HGF, hepatocyte growth factor; IL-1ra, IL-1 receptor antagonist; MUFAs, monounsaturated fatty acids; TOR1AIP1, Torsin-1A-interacting protein 1; UFA, unfavorable adiposity; IL-6, interleukin-6; ALT, alanine aminotransferase; IDO 1, Indoleamine 2,3-dioxygenase 1; PDGF-bb, platelet-derived growth factor BB; SCGF-β, stem cell growth factor-beta; TSH, thyroid-stimulating hormone; β-NGF, beta nerve growth factor; M.VLDL.TG, Triglycerides in medium VLDL; MSP, microseminoprotein-beta; CCB, calcium channel blockers; PPARG, peroxisome proliferator activated receptor γ; PCSK9, proprotein convertase subtilisin/kexin type 9; SLE, systemic lupus erythematosus; IVW, inverse variance weighted; MR, mendelian randomization; OR, odds ratio.

Table). Robust evidence was assigned to PCSK9 inhibition (OR = 0.85, 95% CI: 0.76, 0.96) and hyperthyroidism (increased susceptibility, OR = 0.86, 95% CI: 0.79, 0.94) with relatively small sample size. Probable evidence was assigned to CCB (per SD, OR = 1.22, 95% CI: 1.06, 1.42), atrial fibrillation (increased susceptibility, OR = 0.96, 95% CI: 0.92, 0.99), and systemic lupus erythematosus (SLE) (increased susceptibility, OR = 0.98, 95% CI: 0.97, 0.99). Insufficient evidence was assigned to genetically proxied perturbation of PPARG (per SD, OR = 1.75, 95% CI: 1.07, 2.85). No significant causal association with prostate cancer was found for the following clinical variables, diseases, and treatments including inflammatory bowel disease, Crohn's disease, UC, heart failure, major depressive disorder, systolic blood pressure, diastolic blood pressure, hypothyroidism, schizophrenia, allergic disease, asthma, vitiligo, T2D, and 7 genetically proxied therapeutic inhibition of drug targets.

## Comparison between associations derived from meta-analyses and MR studies

Taking evidence grading results into consideration, no factor showed notable effect on modifying prostate cancer risk with high-quality evidence (Fig 6). In total 26 overlapping factors investigated by both meta-analyses and MR studies were identified, and only 3 factors showed consistent significant associations, yet with no consistent robust evidence: physical activity (PA) (occupational PA in meta: OR = 0.87, 95% CI: 0.80, 0.94, highly suggestive; accelerator-measured PA in MR: OR = 0.49, 95% CI: 0.33, 0.72, insufficient), height (meta: OR = 1.09, 95% CI: 1.06, 1.12, suggestive; MR: OR = 1.07, 95% CI: 1.01, 1.15, insufficient), and smoking (current smoking in meta: OR = 0.74, 95% CI: 0.68, 0.80, suggestive; smoking initiation in MR: OR = 0.91, 95% CI: 0.86, 0.97, robust). Eleven factors including total dairy product, birth weight, calcium, CRP, circulating 25-hydroxyvitamin D, and UC positively linked with prostate cancer and coffee, selenium, vitamin E, schizophrenia, and T2D inversely associated with prostate cancer showed null causal associations by MR studies. However, 3 factors with statistically significant causal associations by MR studies were null in meta-analyses (LDL, zinc, and BMI). Another 9 factors were not significantly associated with prostate cancer neither in meta-analyses nor in MR studies (S8 Table). Except for the overlapping factors, comparison was limited between meta-analyses and MR studies for other factors largely due to unavailability. For example, most of the dietary factors identified in meta-analyses were not suitable for conducting MR studies due to lack of appropriate instrumental variables, whereas some factors found significant in MR studies did not have available meta-analyses (education attainment, morning chronotype, puberty timing, and many biomarkers).

## Discussion

To the best of our knowledge, this large-scale umbrella review conducted a very comprehensive appraisal of the evidence strength of associations between various factors and the risk of developing prostate cancer, based on meta-analyses of prospective observational studies and MR studies. Collectively, 92 meta-analyses and 64 MR studies generated 268 associations with the risk of prostate cancer, covering 6 categories: lifestyle; diet and nutrition; anthropometric

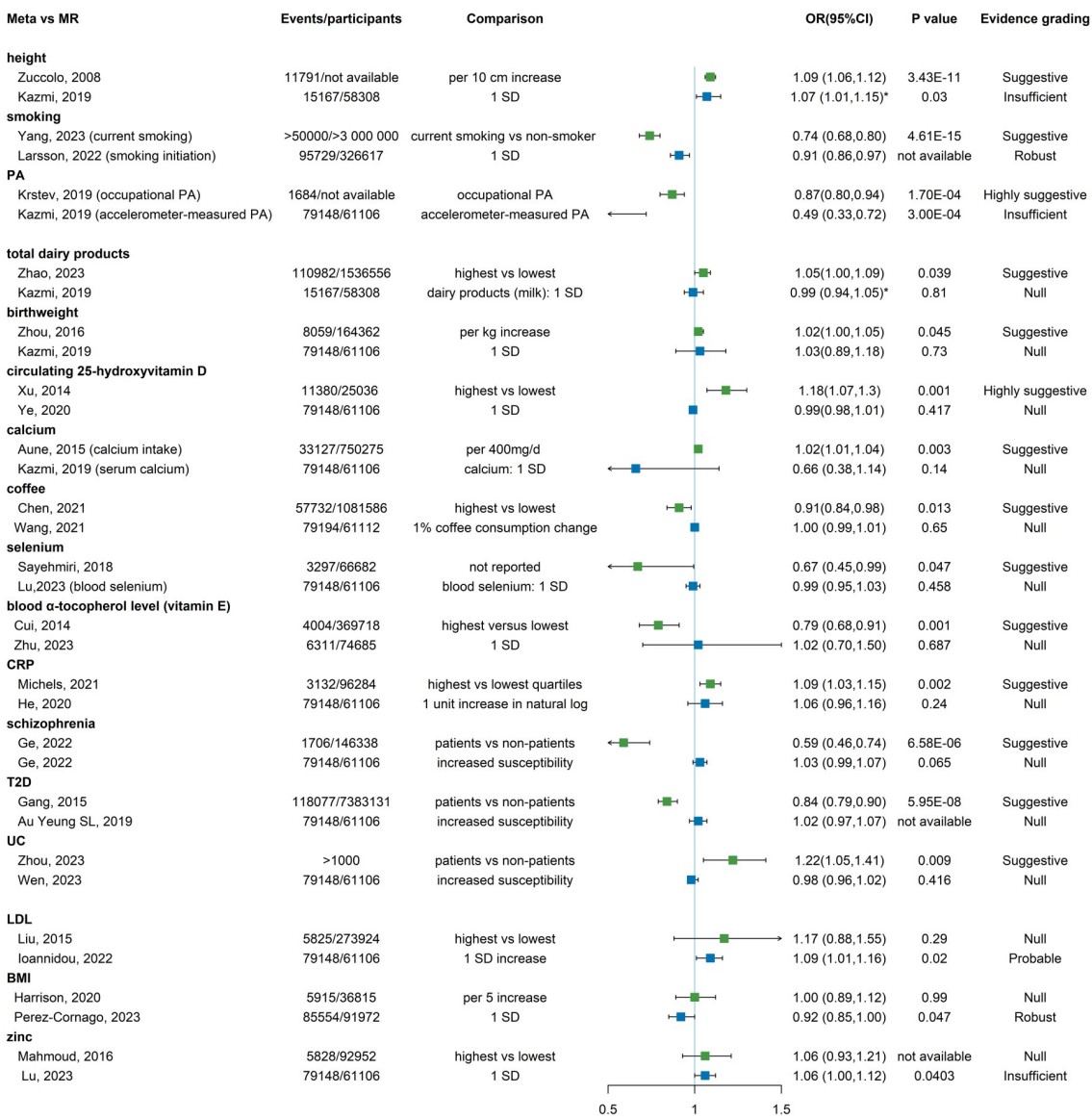

**Fig 6. Comparison between meta-analyses and MR studies.** The statistical test to determine the *P* value in meta-analyses was the random-effects inverse-variance model with DerSimonian—Laird method. The statistical test to determine the P value in MR study was the IVW regression analysis. The effect estimates OR from meta-analyses and MR studies are represented by the green and blue squares, respectively, and 95% CI by the horizontal lines. Metrics with * denoting the outcome was high-grade, aggressive, or advanced prostate cancer. NA, not available; SD, standard deviation; PA, physical activity; CRP, C-reactive protein; T2D, type 2 diabetes; UC, ulcerative colitis; LDL, low-density lipoprotein; BMI, body mass index; IVW, inverse variance weighted; MR, mendelian randomization; OR, odds ratio.

indices; clinical variables, diseases, and treatments; biomarkers; and environmental factors. Further evidence grading on statistically significant associations according to respective pre-specified criteria was performed.

Concerning meta-analyses, our results corroborate largely with previous findings mainly in the category of diet and nutrition [9], including sweetened beverage, vitamin D, folate, dairy product, processed meat, egg consumption increasing the risk of prostate cancer and selenium and soy consumption decreasing the risk. Compared with previous researches, this umbrella

review has the strength of updated evidence and expanded categories of risk factors. The existing umbrella review by Markozannes and colleagues in 2016 [9] was conducted based on literature published up to April 30, 2013, while in our umbrella review all the included meta-analyses for data synthesis were published after 2014 except 2 articles [72,105], of which 57.6% (53/92) were published after 2020, presenting updated evidence for each factor. Secondly, the previous umbrella review studied associations of 23 foods, 31 nutrients, 8 indices of body size and 3 indices of physical activity, while our umbrella review greatly expanded the categories of risk factors by containing 6 categories covering lifestyle; diet and nutrition; anthropometric indices; clinical variables, diseases, and treatments; biomarkers; and environmental factors, bringing the total number of studied factors to 123. Furthermore, this umbrella review collected evidence of clinical variables, diseases, and treatments including preexisting diseases, medication, and surgery, which was often neglected in previous reviews. Diseases such as melanoma, acne in adolescence, UC, infertility, prostatitis, and BPH associated with higher prostate cancer risk indicated shared biological mechanisms such as hormone dependency, inflammation [177], and genetic susceptibility [88,92,97]. We could approach these associations from the perspective of shared causal intermediary pathway or mechanisms to investigate the carcinogenesis of prostate cancer, which warrants further researches such as genetic, functional, and pharmaceutical studies.

Apart from updated evidence and expanded categories, the unique strength of this umbrella review is the comparison between high-quality evidence from meta-analyses of prospective observational studies and MR studies. Integrating epidemiological evidence and MR causal inference, with the former providing the foundation for MR causal exploration while MR helping verify the causality in turn, provides useful insights in examining intrinsic relationships. In this umbrella review, however, the comparison between observational associations by meta-analysis and genetically estimated causality by MR does not provide robust evidence due to the lack of overlapping observations as well as the lack of high-quality evidence, especially in MR studies. First, concerning height, MR analyses on height provided insufficient evidence of its causal association with prostate cancer, in addition to inconsistent results from other identified MR studies [178–180], which is not very supportive of this association. Height is implicated in many biological pathways such as skeletal growth, fibroblast growth factor (FGF) signaling, WNT (Wingless/Integrated) signaling, regulation of β-catenin, mammalian target of rapamycin (mTOR) signaling [181], and associates with overall cancer risk and mortality [178]. A plausible mechanism involves dietary programming of the IGF-1, which plays an important role in the regulation of postnatal growth and is also associated with prepubertal growth in height [182]. Thus, the variations in the IGF-1 system might underlie associations of height with prostate cancers that are more likely to progress [183]. Still the causal mechanism of height in progressive prostate cancer needs further investigation. Smoking, albeit with consistent effect, should be taken prudently for the observed effect was moderate and mixed and that positive association in earlier years (before 1995) and with mortality collectively suggested a link to aggressive prostate cancer rather than indolent one [184]. Current smoking was believed to be associated with a lower likelihood of prostate-specific antigen (PSA) testing, and individuals with a smoking history were less likely to undergo prostate biopsy [185,186]. Consequently, the detection rate of prostate cancer could be relatively lower among participants in the PSA screening era. Another possible explanation is that smoking is the leading risk factor for death among males [187]. Smokers may die from smoking attributable diseases including cancers, cardiovascular diseases, and respiratory diseases before their diagnosis of prostate cancer. In addition, multiple inconsistent exposure categories for smoking such as current smoking, former smoking, and ever smoked, etc., might contribute to the varied results. To sum up, measures should be taken to help smokers to be more compliant with early cancer

screening and to quit smoking [30]. Concerning physical activity, physical activity may be associated with cancer through several pathways related to oxidative stress, DNA methylation, telomere length, immune function, and gut microbiome [188]. Shorter duration aerobic physical activity stimulates short-term increases in immunoglobulins, neutrophils, natural killer cells, cytotoxic T cells, and immature B cells, which over time enhance immunosurveillance [189]. Physical activity reduces adipose tissue and correcting metabolic abnormalities, which has been shown to reduce plasma insulin and increase insulin sensitivity and glucose metabolism, thereby lowering the risk of certain cancers [190]. In terms of cancer progression, physical activity may predispose to biologically less aggressive tumors and may improve functional capacity to tolerate and complete cancer treatment, thereby slowing down cancer progression [191]. The results regarding physical activity in this umbrella review are not robust because accelerator-measured physical activity showed a protective effect on prostate cancer in MR but with very weak instrumental strength explaining only 0.1% of the variance. In addition, in meta-analyses occupational physical activity was graded as highly suggestive evidence but overall physical activity showed null associations with prostate cancer, possibly attributed to differed measurement.

Several limitations should be noted in this umbrella review. First, missing literature may exist despite of exhaustive literature search, and some factors that were not assessed at the meta-analysis level or failed the inclusion criteria may be overlooked. Second, most cohort studies were conducted in developed western countries, and hence findings of this current study are limited mainly for European descendants. Despite subgroup analysis performed by ethnicity, it was greatly limited by sparsity of data on non-white populations. Effects of different risk factors on prostate cancer may vary between ethnicities, which may be attributed to diverse genetic backgrounds and lifestyles. Data on prostate cancer incidence in Asian countries might be statistically biased by the immature implementation of early screening practice and national cancer registry [192]. As prostate cancer is expected to rise in developing countries due to increased aging and popularity of PSA screening, data of non-white population are accumulating and await evaluation. Third, heterogenous effects based on prostate cancer classifications suggest both pathological variation of prostate cancer and diverse effects of exposure factors. For instance, smoking was found to be inversely associated with total prostate cancer, but its effect on aggressive prostate cancer appeared to be the opposite in some literature [193]. Therefore, it is necessary to conduct more precise evaluations on associations with further characterizations considering the complex clinical and pathological nature of prostate cancer. Fourth, evidence grading criteria both for meta-analyses and MR studies could be refined, for example, considering magnitude of effect size and levels of sample size, which requires academically sound innovation and collective effort from the broad science community.

Some implications for next step of research can be derived from this umbrella review. First, the discrepancy that a fair number of factors explored in MR studies are not found in meta-analyses or observational studies should be noted. The accessibility of abundant resources in MR-base may permit analyses to be performed without careful consideration of the epidemiological evidence/background that are being made or the assumptions inherent in the approach [194]. Therefore, it is suggested that MR be performed based on properly and adequately evaluating evidence provided by epidemiological studies. MR results that are not biologically sound or supported by observational studies should be interpreted with caution. Second, the identification of risk factors that are robustly associated with risk of prostate cancer avail targeted prevention strategies. Biomarkers identified in MR studies warrant further investigation, which may benefit future research on prostate cancer carcinogenesis, prevention, and screening. Third, weak and insufficient evidence identified in this umbrella review warrant further investigations.

In summary, this umbrella review provides a comprehensive evaluation on risk factors associated with prostate cancer as well as large-scale comparison between observational associations by meta-analysis and genetically estimated causality by MR analyses. Though no robust association is identified due to the lack of overlapping robust evidence based on existing literature, future researches are warranted to further our understanding on prostate cancer risk.

## Supporting information

**S1 PRISMA Checklist. Prisma 2020 checklist.**
(DOCX)

**S1 Text. Search strategies.**
(DOCX)

**S2 Text. Statistical analysis protocol.**
(DOCX)

**S1 Fig. Forest plots of significant associations in meta-analyses.** The effect estimates are presented as risk ratios (RR) with 95% confidence intervals (95% CI).
(PDF)

**S2 Fig. Forest plots of subgroup analyses according to ethnicity (white versus non-white).** The 2 dashed line indicated the odds ratios derived from the common effect model (the loosely dashed line) and from random-effects model (the densely dashed line), respectively. W, white population; non-W, non-white population; CI, confidence interval.
(PDF)

**S1 Table. Selection of meta-analyses.**
(XLSX)

**S2 Table. Selection of MR studies.** IVs, instrumental variables; OR, odds ratio; CI, confidence interval; BMI, body mass index; IGF, insulin-like growth factor; IGFBP, IGF-binding protein; HDL, high-density lipoprotein; LDL, low-density lipoprotein; TG, triglyceride; CCB, calcium channel blockers; PCSK9, proprotein convertase subtilisin/kexin type 9; SHBG, sex-hormone binding globulin.
(XLSX)

**S3 Table. Details of AMSTAR-2 grading for quality of meta-analyses.** Y: yes (1 point); PY: partial yes (0.5 point); N: no (0 point); * denoting the critical AMSTAR-2 items; critical item score = total score of 0 (N), 0.5 (PY), and 1 (Y) on the critical AMSTAR-2 items; total score = total score of 0 (N), 0.5 (PY), and 1 (Y) on all AMSTAR-2 items.
(DOCX)

**S4 Table. Details of evidence grading for significant associations from meta-analyses.** NA: not available; PI, prediction interval; PA, physical activity; CRP, C-reactive protein; T2D, type 2 diabetes; BPH, benign prostate hyperplasia; UC, ulcerative colitis; HIV, human immunodeficiency virus; AIDS, acquired immune deficiency syndrome.
(DOCX)

**S5 Table. Basic characteristics of included meta-analyses and evidence grading results.** The statistical test to determine the *P* value in meta-analyses was using the random-effects inverse-variance model with DerSimonian—Laird method. Metrics with * denoting advanced, aggressive, high-grade, or lethal prostate cancer, metrics with # denoting nonadvanced,

nonaggressive, or localized prostate cancer. W, White; A, Asian; RR, risk ratio; OR, odds ratio; HR, hazard ratio; SIR, standard incidence ratio; SRRE, summary relative risk estimate; NR, not reported; NA, not available; PA, physical activity; DHA, docosahexaenoic acids; EPA, eicosapentaenoic; HDL, high-density lipoprotein; LDL, low-density lipoprotein; CRP, C-reactive protein; T2D, type 2 diabetes; BPH, benign prostate hyperplasia; HIV, human immunodeficiency virus; AIDS, acquired immune deficiency syndrome; CD, Crohn's disease; UC, ulcerative colitis; AASVs, anti-neutrophil cytoplasm antibody associated vasculitides; ACEI, angiotensin converting enzyme inhibitors; NSAID, nonsteroidal anti-inflammatory drug; CCB, calcium channel blockers.
(DOCX)

**S6 Table. Subgroup analyses according to ethnicity (white versus non-white).** N, number of datasets in the corresponding meta-analysis; OR, odds ratio; CI, confidence interval. Regular use of aspirin: users vs. non-users; Total calcium intake: per 400 mg/d; Coffee: highest vs. lowest; Current smoking: current smoking vs. non-smoker (never smokers plus former smokers); Daidzein: highest vs. lowest; Finasteride: users vs. non-users; Firefighter: ever employment as a career firefighter vs. general population; Height: per 10 cm increase; Soy consumption: highest vs. lowest; Total dairy products: highest vs. lowest; Ulcerative colitis: patients vs. non-patients.
(DOCX)

**S7 Table. Basic characteristics of included MR studies and evidence grading results.** The statistical test to determine the *P* value in MR study was the inverse variance weighted (IVW) regression analysis; [§] denoting the exposure population source was of Asian ancestry or mixed ancestry; [*] denoting the outcome of MR studies was aggressive prostate cancer; [#] denoting the outcome of MR studies was early-onset prostate cancer; [+] denoting the summary metric of this MR study was beta estimates. NA, not available; SD, standard deviation; PRACTICAL: The Prostate Cancer Association Group to Investigate Cancer Associated Alterations in the Genome consortium; PA, physical activity; BMI, body mass index; UFA, unfavorable adiposity; FA, favorable adiposity; HbA1c, hemoglobin A1c; GST, glutathione s-transferase; SOD, superoxide dismutase; CAT, catalase; GPX, glutathione peroxidase; IL, interleukin; IL-1b, IL-1 beta; IL-1ra, IL-1 receptor antagonist; IL-2ra, IL-2 receptor alpha subunit; IL-6ra, IL-6 receptor subunit alpha; ALT, alanine aminotransferase; VEGF, vascular endothelial growth factor; IGF, insulin-like growth factor; IGFBP, IGF-binding protein; TOR1AIP1, Torsin-1A-interacting protein 1; MUFAs, monounsaturated fatty acids; AA, Arachidonic acid; ALA, α -linolenic acid; DHA, Docosahexaenoic acid; DPA, Docosapentaenoic acid; EPA, Eicosapentaenoic acid; LA, linoleic acid; OA, Oleic acid; PA, Palmitic acid; POA, Palmitoleic acid; SA, Stearic acid; CRP, C-reactive protein; HDL, high-density lipoprotein; LDL, low-density lipoprotein; Lp(a), lipoprotein A; TG, triglyceride; apo A, apoprotein A; apo B, apoprotein B; VLDL, very low-density lipoprotein; S.HDL.TG, Triglycerides in small HDL; M.VLDL.TG, Triglycerides in medium VLDL; PDGF-bb, platelet-derived growth factor BB; β-NGF, beta nerve growth factor; SCGF-β, stem cell growth factor-beta; HGF, hepatocyte growth factor; CCL2, Chemokine (C-C motif) ligand 2; CCL4, Chemokine (C-C motif) ligand 4; IDO 1, Indoleamine 2,3-dioxygenase 1; MSP, microseminoprotein-beta; LTL, leukocyte telomere length; SHBG, sex-hormone binding globulin; TSH, thyroid-stimulating hormone; CCB, calcium channel blockers; PCSK9, proprotein convertase subtilisin/kexin type 9; PPARG, peroxisome proliferator activated receptor γ; ABCC8, ATP binding cassette subfamily C member 8; GLP1R, glucagon-like peptide 1 receptor; ACE, angiotensin-converting enzyme; ADRB1, β-1 adrenergic receptor; NCC, sodium-chloride symporter; SBP, systolic blood pressure; DBP, diastolic blood pressure; MDD, major depressive disorder; SLE, systemic lupus erythematosus; IBD, inflammatory

bowel disease; CD, Crohn's disease; UC, ulcerative colitis; T2D, type 2 diabetes; HMG-CoA, 3-hydroxy-3-methylglutaryl coenzyme A; NPC1L1, Niemann-Pick C1-Like 1.
(DOCX)

**S8 Table. Overall comparison between meta-analyses and MR studies.** Metrics with * denoting the outcome was advanced, aggressive, high-grade, or lethal prostate cancer. Other null associations of biomarkers in MR studies were recorded in a previous review by Markozannes and colleagues (reference [19]). PA, physical activity; DHA, docosahexaenoic acids; EPA, eicosapentaenoic; HDL, high-density lipoprotein; LDL, low-density lipoprotein; CRP, C-reactive protein; T2D, type 2 diabetes; BPH, benign prostate hyperplasia; HIV, human immunodeficiency virus; AIDS, acquired immune deficiency syndrome; CD, Crohn's disease; UC, ulcerative colitis; AASVs, anti-neutrophil cytoplasm antibody associated vasculitides; ACEI, angiotensin converting enzyme inhibitors; NSAID, nonsteroidal anti-inflammatory drug; CCB, calcium channel blockers; TG, triglyceride; MUFAs, monounsaturated fatty acids; MDD, major depressive disorder; LTL, leukocyte telomere length; IGF, insulin-like growth factor; IGFBP, IGF-binding protein; TOR1AIP1, Torsin-1A-interacting protein 1; IL-6ra, IL-6 receptor subunit alpha; IDO 1, Indoleamine 2,3-dioxygenase 1; SCGF-β, stem cell growth factor-beta; β-NGF, beta nerve growth factor; MSP, microseminoprotein-beta; ALT, alanine aminotransferase; SLE, systemic lupus erythematosus; TSH, thyroid-stimulating hormone; PCSK9, proprotein convertase subtilisin/kexin type 9; PPARG, peroxisome proliferator activated receptor γ; SHBG, sex-hormone binding globulin; S.HDL.TG, Triglycerides in small HDL; M.VLDL.TG, Triglycerides in medium VLDL; PDGF-bb, platelet-derived growth factor BB.
(DOCX)

## Author Contributions

**Conceptualization:** Xia Jiang, Ben Zhang.

**Data curation:** Huijie Cui, Wenqiang Zhang, Li Zhang, Yang Qu.

**Formal analysis:** Huijie Cui, Wenqiang Zhang, Li Zhang.

**Funding acquisition:** Ling Zhang, Yanfang Yang, Yuqin Yao, Jiayuan Li, Zhenmi Liu, Chunxia Yang, Ben Zhang.

**Investigation:** Huijie Cui, Wenqiang Zhang, Li Zhang, Yang Qu.

**Resources:** Ben Zhang.

**Software:** Zhengxing Xu, Zhixin Tan, Peijing Yan, Mingshuang Tang, Chao Yang, Yutong Wang, Lin Chen, Chenghan Xiao, Yanqiu Zou, Yunjie Liu.

**Supervision:** Ling Zhang, Yanfang Yang, Yuqin Yao, Jiayuan Li, Zhenmi Liu, Chunxia Yang, Xia Jiang, Ben Zhang.

**Visualization:** Zhengxing Xu, Zhixin Tan, Peijing Yan, Mingshuang Tang, Chao Yang, Yutong Wang, Lin Chen, Chenghan Xiao, Yanqiu Zou, Yunjie Liu.

**Writing – original draft:** Huijie Cui, Wenqiang Zhang, Li Zhang.

**Writing – review & editing:** Huijie Cui, Wenqiang Zhang, Li Zhang, Yang Qu, Zhengxing Xu, Zhixin Tan, Peijing Yan, Mingshuang Tang, Chao Yang, Yutong Wang, Lin Chen, Chenghan Xiao, Yanqiu Zou, Yunjie Liu, Xia Jiang, Ben Zhang.

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
