## [Editor Report · Decision Letter 0]

21 Jun 2023

Dear Dr Zhang, 

Thank you for submitting your manuscript entitled "Understanding the risk of prostate cancer: an umbrella review of observational and genetic evidence" for consideration by PLOS Medicine.

Your manuscript has now been evaluated by the PLOS Medicine editorial staff and I am writing to let you know that we would like to send your submission out for external peer review.

Please re-submit your manuscript within two working days, i.e. by Jun 23 2023 11:59PM.

Kind regards,

Louise Gaynor-Brook, MBBS PhD

Senior Editor

PLOS Medicine

---

## [Decision Letter · Decision Letter 1]

6 Sep 2023

Dear Dr. Zhang,

Thank you very much for submitting your manuscript "Understanding the risk of prostate cancer: an umbrella review of observational and genetic evidence" (PMEDICINE-D-23-01730R1) for consideration at PLOS Medicine. 

Your paper was evaluated by a senior editor and discussed among all the editors here. It was also discussed with an academic editor with relevant expertise, and sent to three independent reviewers, including a statistical reviewer. The reviews are appended at the bottom of this email and any accompanying reviewer attachments can be seen via the link below:

[LINK]

In light of these reviews, I am afraid that we will not be able to accept the manuscript for publication in the journal in its current form, but we would like to consider a revised version that addresses the reviewers' and editors' comments. Obviously we cannot make any decision about publication until we have seen the revised manuscript and your response, and we plan to seek re-review by one or more of the reviewers. 

We expect to receive your revised manuscript by Sep 27 2023 11:59PM. Please email us (plosmedicine@plos.org) if you have any questions or concerns.

We look forward to receiving your revised manuscript. 

Sincerely,

Louise Gaynor-Brook, MBBS PhD

plosmedicine.org

General comments:

Please include line numbers in your revised manuscript, ideally not starting from 1 with each new page.

Please consider having your manuscript reviewed by someone with full professional proficiency in English.

Please replace "Caucasian" with "white" throughout the paper.

Please revise ‘sweeten beverage’ to ‘sweetened beverage’

Throughout the paper, please adapt reference call-outs to the following style: "... every year [1,2]." (noting the absence of spaces within the square brackets).

Title: Please revise your title according to PLOS Medicine's style. We suggest “Risk factors for prostate cancer: An umbrella review of evidence from observational studies and Mendelian Randomization studies” or similar

Abstract:

Please report your abstract according to PRISMA for abstracts, following the PLOS Medicine abstract structure (Background, Methods and Findings, Conclusions) http://www.plosmedicine.org/article/info:doi/10.1371/journal.pmed.1001419

Abstract Background: Please expand on the context of why the study is important. 

Please revise ‘rising’ for clarity; such as ‘The incidence of prostate cancer is increasing in older males’

Please remove ‘perplexing’

Please remove ‘aging elder’

Please remove “apart from race, age, and heredity”, which are not modifiable risk factors. 

The final sentence should clearly state the study question.

Please revise to ‘prospective observational studies’

Please revise to ‘Mendelian’

Abstract Methods and Findings:

Please provide the dates of search (start and end), number of studies included, eligibility criteria, and synthesis/appraisal methods. 

Please quantify the main results presented in the abstract

Please revise ‘In total, 94 associations…’

In the last sentence of the Abstract Methods and Findings section, please describe 2-3 of the main limitations of the study's methodology.

Abstract Conclusions:

Please begin your Abstract Conclusions with "In this study, we observed ..." or similar, to summarize the main findings from your study, without overstating your conclusions. Please emphasize what is new and address the implications of your study, being careful to avoid assertions of primacy. Please avoid vague statements such as "significant relevance for public health". 

Author Summary:

In the final bullet point of ‘What Do These Findings Mean?’, please describe the main limitations of the study in non-technical language.

Introduction:

Please temper assertions of primacy by adding ‘to the best of our knowledge’ or similar, in the following sentences:

“...remain unevaluated by umbrella reviews.” 

“...have rarely been compared…” 

“...accumulating new and unevaluated…”

“...sheds novel lights on PrCa oncogenesis…”

Please remove “bears significant public health and clinical relevance.”

Please conclude the Introduction with a clear description of the study question or hypothesis.

Methods:

Did your study have a prospective protocol or analysis plan? Please state this (either way) early in the Methods section. If a prospective analysis plan was used in designing the study, please include the relevant prospectively written document with your revised manuscript as a Supporting Information file to be published alongside your study, and cite it in the Methods section. If no such document exists, please make sure that the Methods section transparently describes when analyses were planned, and if/when reported analyses differed from those that were planned. Changes in the analysis-- including those made in response to peer review comments-- should be identified as such in the Methods section of the paper, with rationale. If a reported analysis was performed based on an interesting but unanticipated pattern in the data, please be clear that the analysis was data-driven.

Please add the following statement, or similar, to the Methods: "This study is reported as per the Preferred Reporting Items for Systematic Reviews and Meta-Analyses (PRISMA) guideline (S1 Checklist)." 

We require that SRs are updated to within roughly 6 months of the expected publication date. Please update your search to the present time.

Please consider including non-English language sources of studies.

Please refer to specific files in your supplementary information e.g. S1 Table, S1 Figure, etc. including for the search strategy

Results: 

Please define all abbreviations at first use e.g. OR, RR, DHA, EPA, BMI, T2D, etc

Where ORs are presented, please specify the comparison group.

Regarding birth weight (OR=1.02, 95% CI: 1.00-1.05) and fat mass (OR=0.88, 95% CI: 0.76-1.00): Please clarify why these are considered significant since they contain the null value

“well documented in the previous review” - please provide a citation

In the ‘lifestyle’ paragraph you state that “Null associations were found between PrCa and 11 lifestyle factors: … physical activity…” but later (on page 26) you state that “physical activity (evidence level: convincing in meta-analysis /insufficient in MR)” - please clarify.

Discussion:

Please present and organize the Discussion as follows: a short, clear summary of the article's findings; what the study adds to existing research and where and why the results may differ from previous research; strengths and limitations of the study; implications and next steps for research, clinical practice, and/or public policy; one-paragraph conclusion.

Please remove all subheadings within your Discussion e.g. Meta-analyses

Please temper assertions of primacy by adding ‘to the best of our knowledge’ or similar e.g. “the hitherto most comprehensive appraisal “ etc 

As before, please clarify your classification in the Discussion of physical activity as convincing/highly suggestive when this was not the result presented in the Results.

“four robust associations in MR” - please note that only 3 associations are listed

Please revise “...confirming the roles of…”, “protective effects of”, “highlighted the role of”, “Higher PrCa incidence was found in”, “provides novel prospects”, “the most comprehensive one”, “providing novel insights“ etc. to avoid use of causal language

Figures:

Please define abbreviations used in the figure legend of each individual figure.

Fig 4: When a p value is given, please specify the statistical test used to determine it in the figure legend.

Tables:

Please define all abbreviations used in the table legend of each individual table.

Supplementary Table 1 - please indicate which exposure is being investigated for each of the sections e.g. Burton 2021 is the most recent of three papers, but what was the exposure of interest in these three? Please also indicate why some names are emboldened, as this doesn’t always appear to correspond to the paper used in your review, and the meaning of the square root symbol / “latest from MR”.

Tables 2 & 3: When a p value is given, please specify the statistical test used to determine it in the table legend.

References:

Please ensure that journal name abbreviations match those found in the National Center for Biotechnology Information (NCBI) databases (http://www.ncbi.nlm.nih.gov/nlmcatalog/journals), and are appropriately formatted and capitalised.

Please also see https://journals.plos.org/plosmedicine/s/submission-guidelines#loc-references for further details on reference formatting. 

Where website addresses are cited, please specify the date of access. 

Supplementary files: 

Please provide titles and legends for each individual table and figure in the Supporting Information.

Please see https://journals.plos.org/plosmedicine/s/supporting-information for our supporting information guidelines. 

Comments from the reviewers:

Reviewer #1: This is a well-conducted umbrella review on the factors modifying prostate cancer risk reported in meta-analyses of prospective studies as well as mendelian randomization studies. The study design, datasets, statistical methods and analyses, and presentation (tables and figures) and interpretation of the results are mostly adequate and of a good standard. However, still a couple of issues needing attention.

1) Table 2 summarises the 72 meta-analyses included in the review. We can see quite a few studies including both Caucasian and Asian cohorts. As it's known the race plays a big role in prostate cancer, it would be very useful if authors could do some subgroup analyses to show whether the claimed risk factors differ, at least between Caucasian and Asian cohorts. I am aware the authors have aknowledged this in limitations citing there are not enough data on race but it seems that data on both Caucasian and Asian are avaiable so it might be possible to conduct a subgroup analysis on this.

2) Figure 3 is very informative. The evidence grading is very useful. However, it would be good to add the I-squared value for heterogeneity too in the figure as it's a part of routine meta-analysis and familar to readers and easy to follow. The Odds ratio, I-squared and evidence grading would make a comprehensive presentation of the results.

Reviewer #2: Cui et al. have put together a comprehensive umbrella review of risk factors associated with prostate cancer via meta-analyses of prospective observational studies and Mendelian randomization studies. 

Major comments 

1. There is some lack of novelty given two umbrella reviews published by Markozannes, which overlap with the work here (2016 European Journal of Cancer, 2022 BMC Medicine). Still, the current work appears comprehensive and might still be additive to the literature.

2. Authors should add information on the exclusion criteria for meta-analyses and Mendelian randomization studies in the Methods section. 

3. In meta-analyses, among the dietary and nutritional factors, authors report that circulating high levels of 25-hydroxyvitamin D associated with PrCa risk. Was the cut-off level defining high or low was same in the meta-analysis review? 

4. Also in meta-analyses and in MR studies, smoking was coming out to be a significant protective factor among lifestyle factors which is strange. Authors should add a paragraph in discussion section about this. 

5. Among the environment factors, authors did not find a significant association of pesticides with PrCa in meta-analysis review which is not in line with the literature (PMID: 27244877). 

6. When comparing association between overlapping factors between meta-analyses and MR studies, only three factors showed consistently significant associations: height, smoking, and physical activity. How do these associations compare with the other umbrella reviews on prostate cancer, i.e., Markozannes et al. European Journal of Cancer, 2016?

Reviewer #3: In their manuscript, entitled "Understanding the risk of prostate cancer: an umbrella review of observational and genetic evidence," authors Cui et al. attempt to make sense of the large collection of environmental exposure associations with prostate cancer. To do so, they conducted a search of all meta-analyses of prospective studies from 1990-2022 published in the English language, assigning evidence grades (convincing, highly suggestive, suggestive, or weak) to correlations detected. Grades reflected on sample size, P-values and 95% confidence intervals, Funnel plot asymmetry, etc. Mendelian randomization studies were graded as robust, probable, suggestive, and insufficient considering P-values and concordance of effect directions.

Findings were: (i) selection of 72 meta-analyses reporting 94 associations (from 360 or so identified studies), predominantly sampling Caucasians from Europe and North America, (ii) 41 significant associations subjected to grade assignment, with only 12 associations showing a 95% confidence interval that excluded a null association, (iii) significant correlations between diet and nutrition and prostate cancer for 11 factors, with highly suggestive evidence for sweetened beverages and circulation vitamin D, (iv) convincing evidence for reduced physical activity and prostate cancer, with suggestive evidence for associations of birth weight, height, and fat mass and prostate cancer, (v) only suggestive evidence of disease impact from medical conditions and/or medications, (vi) suggestive evidence for C-reactive protein levels and prostate cancer risk, (vii) highly suggestive evidence for asbestos exposure and disease risk, (viii) 18 associations discriminated from Mendelian randomization studies, (ix) robust evidence only for circulating phosphate, leukocyte telomere length, and PCSK9 levels, and (x) poor concordance between meta-analysis findings and Menedelian randomization results.

With these data and analyses, the authors could confidently proffer only a limited amount of associations that were convincing (meta-analyses) or robust (Mendelian randomization). As such, a reasonable treatment of confounding was provided. 

The major challenge for all of the studies is the case definition: even though autopsies of men not known to have the disease reveal prostate cancer in more than a third of men as the age (see Jahn JL et al. Int J Cancer 137:2795-2802, 2015 and Bell KJL et al. Int J Cancer 157: 1749-1757, 2015), most of those cancers are not diagnosed. Most prostate cancers in the US and Europe are now screen-detected, so disease diagnosis inevitably reflects the likelihood that someone has been screened (and diagnosed) rather than the propensity to have suffered the onset of prostate cancer. Some analyses have attempted to circumvent this challenge by focusing of Gleason grade/score or on prostate cancer mortality. Were there any attempts by the current folks to do so?

Another approach has been to limit analyses to cohorts where men underwent prostate biopsy as part of the study (versus 'for-cause'), such as in the Prostate Cancer Prevention Trial or the Selenium and Vitamin E Cancer Prevention Trial. Were data from these trials (one was mentioned in the manuscript text) examined separately?

[LINK]

---

## [Decision Letter · Decision Letter 2]

11 Dec 2023

Dear Dr. Zhang,

Thank you very much for submitting your manuscript "Risk factors for prostate cancer: An umbrella review of evidence from prospective observational studies and Mendelian randomization studies" (PMEDICINE-D-23-01730R2) for consideration at PLOS Medicine. 

Your paper was re-evaluated by three independent reviewers, including a statistical reviewer, and discussed among all the editors here. It was also discussed with an academic editor with relevant expertise. The reviews are appended at the bottom of this email and any accompanying reviewer attachments can be seen via the link below:

[LINK]

I am pleased to invite you to submit a revised version of your manuscript that addresses the reviewers' and editors' comments in full. Further to our recent correspondence by email, it is essential that the Methods section is revised to detail how meta-analyses were repeated in your review, and that assessment of quality is reconducted using a more appropriate tool. We cannot make any decision about publication until we have seen the revised manuscript and your response, and we plan to seek re-review by one or more of the reviewers. 

We expect to receive your revised manuscript by Jan 01 2024 11:59PM. Please email me (lgaynor@plos.org) if you have any questions or concerns.

We look forward to receiving your revised manuscript. 

Sincerely,

Louise Gaynor-Brook, MBBS PhD

plosmedicine.org

We have identified a BMJ Medicine guideline on umbrella reviews (http://dx.doi.org/10.1136/bmjmed-2021-000071) which states that "Researchers should use the study specific data extracted from each SRMA to repeat each meta-analysis separately rather than report the meta-analytical result as presented in the original SRMA." It is not clear in your Methods section whether meta-analyses were repeated in your umbrella review. Please ensure that it is clearly states in your Methods whether meta-analyses were repeated. From lines 163-182, it appears that this has only been partially done, when a mix of observational studies were included in the original MA, and where studies do not provide "required information". 

We note that you have referenced the original 11-item AMSTAR tool for assessing MAs of randomised studies. Since your umbrella review only includes non-randomised studies, please use another tool that is valid for non-randomised studies specifically. AMSTAR-2 would be acceptable if a tool designed specifically for assessing quality of non-randomised studies cannot be identified. 

Comments from the reviewers:

Reviewer #1: Thanks authors for their effort to improve the manuscript. I am mostly satisfied with the response and revision. However, the extra subgroup analysis results on white vs non-white HAVE NOT been included in S8 Table as the authors claimed

"Additionally, subgroup analyses between whites and non-whites population were performed upon request. The subgroup analyses results showed the effects remained largely consistent for white cohorts (S8 Table) and that number of researches on non-white population was very limited"

This is inadequate. Can authors make sure this subgroup analysis results are included somewhere in the paper?

Reviewer #2: The authors have done a nice job addressing my reviewer comments. My remaining comments are as follows:

1. Only 3 factors showed consistent significant associations with prostate cancer in both meta-analyses and MR studies. Of these, smoking does not make sense, and the association between physical activity was not robust in MR accelerator-measured physical activity. Thus, height seems to be the only real positive factor that came out of this large umbrella review, which adds little novelty to the existing literature.

2. The Table 6 subgroup analysis between white and non-white populations is barely mentioned in the Results, rather an off-hand comment is made that this was done to address reviewer comments. It is also not mentioned at all in the Discussion. Can these results be expanded upon in the results and also discussed with a paragraph in the discussion? For example, the association between dairy products and prostate cancer risk seems much greater in non-white populations (although this appears to be the result of a single study).

3. Lines 262 through 265 in the Results should depict the actual numbers and percentages rather than just approximations. For example, "The study design contained mostly prospective cohort, with a small portion (less than 10%) of nested control study and case-cohort study". What were the exact percentages of nested control studies and case-cohort studies, and how many such studies were included? How many studies were prospective? (Number and percentage).

[LINK]

---

## [Editor Report · Decision Letter 3]

23 Jan 2024

Dear Dr. Zhang,

Thank you very much for re-submitting your manuscript "Risk factors for prostate cancer: An umbrella review of prospective observational studies and Mendelian randomization analyses" (PMEDICINE-D-23-01730R3) for review by PLOS Medicine.

I appreciate your detailed response to the editors' and reviewers' comments. I have discussed the paper with my colleagues and the academic editor and it has also been seen again by one of the original reviewers. The changes made to the paper were satisfactory to the reviewer. However, there are still some remaining comments that need to be addressed. As such, we intend to accept the paper for publication, pending your attention to the editorial comments below in a further revision. When submitting your revised paper, please once again include a detailed point-by-point response to the editorial comments.

[LINK]

In revising the manuscript for further consideration here, please ensure you address the specific points made by each reviewer and the editors. In your rebuttal letter you should indicate your response to the reviewers' and editors' comments and the changes you have made in the manuscript. Please submit a clean version of the paper as the main article file. A version with changes marked must also be uploaded as a marked up manuscript file. Please also check the guidelines for revised papers at http://journals.plos.org/plosmedicine/s/revising-your-manuscript for any that apply to your paper.

We ask that you submit your revision within 1 week (Jan 30 2024). However, if this deadline is not feasible, please contact me by email, and we can discuss a suitable alternative.

Please do not hesitate to contact me directly with any questions (aschaefer@plos.org). If you reply directly to this message, please be sure to 'Reply All' so your message comes directly to my inbox.

We look forward to receiving the revised manuscript.

Sincerely,

Alexandra Schaefer, PhD

On behalf of:

Louise Gaynor-Brook, MBBS PhD

Senior Editor 

PLOS Medicine

Requests from Editors:

GENERAL

1) Please replace hyphens with commas when reporting 95% CI values. The use of commas to separate upper and lower bounds, as opposed to hyphens, could be confused with reporting negative values.

2) We noted that the section on data extraction and synthesis for meta-analyses is very detailed, whereas this section for MR studies is rather short. Please revise to ensure that sufficient detail is provided for both approaches.

3) Please provide the initials of the authors who searched the literature, assessed quality etc. 

4) We noticed that throughout the Results section, none of the included meta-analyses are referenced when discussing the relevant data. Please revise.

ABSTRACT

1) l.28: “The objective is to identify…” – please change to “The objective of this study was to identify…”

2) l.46: Please clarify what you mean by “outdated studies”.

INTRODUCTION

l.107: “The earliest umbrella review on prostate cancer..” – please add “to our knowledge” or similar.

METHODS AND RESULTS

Please ensure that the manuscript is revised to improve clarity and conciseness. In particular, we feel that the presentation of values in the Results section is not detailed enough, and we have noted several instances where numbers do not add up.

1) l.222: Please change to “loosened effect P value threshold of <0.001”.

2) ll.228-229, please change to: Also, associations showing the presence of both large heterogeneity and high publication bias (I2≥50% and Egger’s P ≤0.1) would be graded as weak.

3) ll.258-260: The numbers provided for the individual six major categories do not add up to 123 observational associations, but to 122. Please check carefully and revise.

4) ll.261-262: Please not only report upper and lower values, but a median with the interquartile range.

5) ll.261-264: When presenting the percentages of the various study designs included in the meta-analysis, it would be helpful to include the total number, i.e. the denominator, so that the calculation is understandable (e.g., “The study design contained mostly cohort studies (1,342 [95.7%] of XXXX)...”.)

6) ll.265-266. Please not only report the median, but also the 25th and 75th percentiles of the data, i.e. the interquartile range.

7) ll.266-271: Please not only report percentages, but also numerator and denominator in parentheses. 

8) l.274: For easy comprehension, we suggest repeating the total number of associations here. For example: “In total 46 (of 123) significant associations…”

9) l.274: We think it would be useful to mention the number of meta-analyses from which the 46 significant associations were derived.

10) ll.283-285: Again, we think it would be useful to mention the number of meta-analyses from which the 11 significant associations were derived.

11) l.285: “(mostly less than 2)” seems to be a vague statement. Could you quantify this?

12) l.289: Please reference the one study and remove the word “single”.

13) l.290 and ongoing: When presenting data, be sure to reference the appropriate graph and/or table.

14) l.291 and ongoing: For clarity, please be sure to present the denominator when applicable. For example: “In total 17 lifestyle factors (of 123 total observations) were identified…”. Please revise throughout the entire main manuscript.

15) l.338: Please change to “…with almost half..”.

16) l.338: We feel the term “medical condition factors” might not be appropriate for all factors included in this group, such as “sulfonylureas” or “regular use of aspirin”. We suggest changing this category to “clinical variables, treatments/procedures, and risk factors” or similar.

17) ll.387 and ongoing: Please specify what “most” and “some” means, i.e. providing actual numbers.

18) ll.391-392: Please revise “There were 94 reported associations (65%) had sensitivity analysis.”. 

19) l.395 and ongoing: Similar to the presentation of "Results of meta-analyses in categories", we suggest that under "Results of MR studies in categories" you should start each paragraph with the number of factors in each category. For example: l.390 " Concerning lifestyle factors (10/145)..."

20) l.396: Under “lifestyle factors” you only discuss 9 out 10 factors. What about the 10th factor? 

21) l.407: Under “anthropometric indices” you only discuss 8 out 9 factors. What about the 9th factor? 

22) ll.418-419: What about the other 57 biomarkers? Please make sure that all factors are discussed, not all in detail, but at least as "remaining factors". Please revise throughout the main manuscript.

23) l.420: Please see the comments about “medical condition factors” and revise accordingly (also in the Discussion section).

DISCUSSION

1) l.457: “more than 200” – please present the exact value.

2) l.477: “These diseases..” – please specify.

3) l.448: The evidence grading for height in MR studies was ‘insufficient’. We feel that the weak evidence in the MR studies is not very supportive of this association, please discuss it. Also, please discuss in more detail the existing literature on how height is implicated in cancer risk/mortality.

4) ll.490-491: Please define ‘FGF’, ‘WNT’ and ‘mTOR’.

5) l.505: Please discuss in more detail the existing literature on how physical activity is implicated in cancer risk/mortality.

6) ll.544-547: We feel that the conclusion section is rather general and does not discuss that the comparison between observational associations by meta-analysis and genetically estimated causality by MR does not provide robust evidence due to the lack of overlapping observations as well as the lack of high-quality evidence, especially in MR studies. Please revise.

FIGURES AND TABLES

1) Table 1: Please change 'Egger P value' to 'Egger's P value'. We also suggest that the result of 'Egger's p-value' be consistently described as 'publication bias' instead of switching between 'publication bias' and 'small study effects'.

2) Figure 1: Please define what “outdated” means.

3) Figure 2: Please mention in the figure description the total number of meta-analyses included. Please consider avoiding the use of red and green in order to make your figure more accessible to those with colour blindness. 

4) Figure 3: Please in the figure description, define the numerical value presented in the two graphs. For example: Numbers presented in the graphs are Odds Ratio with 95% confidence intervals. Also: “¶ represents significant associations” – does significance equal p<0.05? Please revise.

REFERENCES

Please thoroughly revise all references and ensure that journal name abbreviations match those found in the National Center for Biotechnology Information (NCBI) databases (http://www.ncbi.nlm.nih.gov/nlmcatalog/journals), and are appropriately formatted and capitalised (e.g., for reference [3] Annals of oncology: official journal of the European Society for Medical Oncology should be Ann Oncol)

SUPPLEMENTARY MATERIAL

1) S1 Fig: Please add in the figure description, that values are presented as Risk Ratios (RR) with 95% confidence intervals (95% CI). Also, please choose a proper heading for each graph (e.g. “egg” does not seem appropriate, better: Egg consumption) and we suggest adding parenthesis the category of each factor (lifestyle, biomarker etc.).

2) S2 Fig: Please change the title of the figure to: Forest plots of subgroup analyses according to ethnicity (white versus non-white). Please define all abbreviations used in the graphs (TE, seTE, CI etc.). Also, please define the meaning of the dashed lines.

3) S6 Table: Please in the footnotes, be sure to add details about the exposure listed. For example: Firefighter: ever employment as a career firefighter vs general population.

4) S7 Table: Please check whether footnote a is correctly formatted as footnote in the table.

SOCIAL MEDIA

To help us extend the reach of your research, please provide any X (formerly known as Twitter) handle(s) that would be appropriate to tag, including your own, your coauthors’, your institution, funder, or lab. Please respond to this email with any handles you wish to be included when we tweet this paper.

Comments from Reviewers:

[LINK]

General Editorial Requests

---

## [Editor Report · Decision Letter 4]

5 Feb 2024

Dear Dr. Zhang,

Thank you very much for re-submitting your manuscript "Risk factors for prostate cancer: An umbrella review of prospective observational studies and Mendelian randomization analyses" (PMEDICINE-D-23-01730R4) for review by PLOS Medicine.

I appreciate your detailed response to the editors' comments. I have discussed the paper with my colleagues and the changes made to the paper were mostly satisfactory to us. However, there are still some remaining comments that will need to be addressed in one final revision. When submitting your revised paper, please once again include a detailed point-by-point response to the editorial comments.

[LINK]

In revising the manuscript for further consideration here, please ensure you address the specific points made by each reviewer and the editors. In your rebuttal letter you should indicate your response to the reviewers' and editors' comments and the changes you have made in the manuscript. Please submit a clean version of the paper as the main article file. A version with changes marked must also be uploaded as a marked up manuscript file. Please also check the guidelines for revised papers at http://journals.plos.org/plosmedicine/s/revising-your-manuscript for any that apply to your paper. 

We ask that you submit your revision within 1 week (Feb 12 2024). However, if this deadline is not feasible, please contact me by email, and we can discuss a suitable alternative.

Please do not hesitate to contact me directly with any questions (aschaefer@plos.org). If you reply directly to this message, please be sure to 'Reply All' so your message comes directly to my inbox.

We look forward to receiving the revised manuscript.

Sincerely,

Alexandra Schaefer, PhD

On behalf of:

Louise Gaynor-Brook, MBBS PhD

Senior Editor 

PLOS Medicine

Requests from Editors:

1) Abstract conclusion: You state that height, physical activity, and smoking had consistently significant effect on the risk of prostate cancer between meta-analyses and MR studies which we think might be misleading as an overall conclusion. Rather, we feel that the comparison between observational associations by meta-analysis and genetically estimated causality by MR does not provide robust evidence due to the lack of overlapping observations as well as the lack of high-quality evidence. Please revise this statement. Editorial suggestion: In this umbrella review, we summarized the associations of various factors with prostate cancer risk and provided comparisons between observational associations by meta-analysis and genetically estimated causality by MR analyses. In the absence of overlapping robust evidence based on the existing literature, no robust associations were identified, but some effects were observed for height, physical activity, and smoking.

2) ll.231-233: Please change to: “Highly suggestive evidence, with the largest component study requirement removed, required a loosened effect P value threshold of <0.001…” (the previous comment was regarding the less-than sign).

3) l.228: The cut-off values for small/large heterogeneity and publication bias need to be revised so that the values of 50% (for heterogeneity) and p=0.1 (for Eggers) are not excluded. For example, no or small heterogeneity should be defined as I²≤50%, or large heterogeneity should be defined as I²≥50%. The same applies to publication bias, i.e. no publication bias should be defined as Egger’s P≥0.1 or publication bias as P≤0.1. Please revise in the text and Table 1.

4) “The 123 observational associations included both the inverse association of finasteride with total prostate cancer and the positive association of finasteride with advanced prostate cancer, which we deemed as two different associations for distinction. Therefore, we counted the total number of associations as 123 though there were 122 factors.” – please briefly include this information in the main text. We suggest you insert it following line 273.

5) ll.304-305, please change to: “there was only one study of five factors, two studies of 4 factors, 3 for ulcerative colitis, and 4 for current smoking on non-white populations.” Please insert references to these studies.

6) Please revise references [18], [26], [29], [124], [126], [152], [162], [166], [169], [189], 

 – the text following the PMICD number should be removed. Please carefully check all references for their correct formatting.

7) Figure 2: Please add to the figure description the definitions of ‘Yes’, ‘Partial Yes’ and ‘No’ as done in S3 Table.

8) Figure 3: In the figure description, please add the definition of the asterisk in graph (A) and the hash in graph (B). 

9) Figure 6: Please define ‘NA’ under abbreviations.

10) S1 Fig/S2 Fig: It seems for some of the graphs bits of text and numbers are overlapping. For example, for the first graph in S1 Fig (Regular use of aspirin), it is following the test for subgroup differences (fixed effects). Please carefully check so that the graphs are properly displayed. Also, please define the meaning of the differently dashed lines (densely and loosely dashed). Please note that “left dashed line” and “right dashed line” are not appropriate definitions. Please add the definitions of abbreviations to both figure descriptions.

11) Please note that the file for S3 Statistical Analysis Protocol was not provided.

Comments from Reviewers:

[LINK]

General Editorial Requests

---

## [Editor Report · Decision Letter 5]

16 Feb 2024

Dear Dr Zhang, 

On behalf of my colleagues and the Academic Editor, Aadel A Chaudhuri, I am pleased to inform you that we have agreed to publish your manuscript "Risk factors for prostate cancer: An umbrella review of prospective observational studies and Mendelian randomization analyses" (PMEDICINE-D-23-01730R5) in PLOS Medicine.

I appreciate your thorough responses to the reviewers' and editors' comments throughout the editorial process. We look forward to publishing your manuscript, and editorially there are only two remaining minor stylistic/presentation points that should be addressed prior to publication. We will carefully check whether the changes have been made. If you have any questions or concerns regarding these final requests, please feel free to contact me at aschaefer@plos.org.

Please see below the minor points that we request you respond to:

1) l.191: Please change "risk ration (RR)" to "risk ratio (RR)".

2) ll.315-316: "there was only one study for five factors [41, 46, 47, 73, 119], two studies for 4 factors [36, 50, 58, 101], 3 for ulcerative colitis [90], and 4 for current smoking [31] on non-white populations." – The numbers of studies mentioned does not match the number of references included here. For example: If there’s only one study for five factors, only one study should be referenced. Please revise.

PRESS

Sincerely, 

Alexandra Schaefer, PhD

On behalf of:

Louise Gaynor-Brook, MBBS PhD 

Senior Editor 

PLOS Medicine